

# Impact of hygroscopic seeding on the initiation of precipitation formation: results of a hybrid bin microphysics parcel model

Istvan Geresdi[1], Lulin Xue[2,3*], Sisi Chen[2], Youssef Wehbe[4], Roelof Bruintjes[2], Jared A. Lee[2], Roy M. Rasmussen[2], Wojciech W. Grabowski[2], Noemi Sarkadi[1], Sarah A. Tessendorf[2]

[1]University of Pécs, Faculty of Science Pécs, Hungary
[2]National Center for Atmospheric Research, Boulder, USA
[3]Hua Xin Chuang Zhi Sci. & Tech. LLC, Beijing, China
[4]National Center of Meteorology, Abu Dhabi 4815, UAE

*Correspondence to*: Lulin Xue (lulin.xue@gmail.com, xuel@ucar.edu)

**Abstract.** A hybrid bin microphysical scheme is developed in a parcel model framework to study how natural aerosol particles and different types of hygroscopic seeding materials affect the precipitation formation. A novel parameter is introduced to describe the impact of different seeding particles on the evolution of the drop size distribution. The results of more than 100 numerical experiments using the hybrid bin parcel model show that: (a) The Ostwald-ripening effect has a substantial contribution to the broadening of the drop size distribution near the cloud base. The efficiency of this effect increases as the updraft velocity decreases. (b) The efficiency of hygroscopic seeding is significant only if the size of the seeding particles is in the coarse particle size range. The presence of the water-soluble background coarse particles reduces the efficiency of the seeding. (c) The efficient broadening of the size distribution due to the seeding depends on the width of the size distribution of water drops in the control cases, but the relation is not as straightforward as in the case of the glaciogenic seeding.

## 1 Introduction

The concept of hygroscopic seeding is based on the hypothesis that the hygroscopic seeding accelerates precipitation formation by suppressing the activation of the natural, less hygroscopic particles, and/or by promoting the formation of larger water drops directly (Mather et al., 1997, Cooper et al., 1997). The seeding particles become activated at lower supersaturation due to their higher hygroscopicity and larger size compared to background cloud condensation nuclei (CCN). The drops formed on the seeding particles grow quickly even under the cloud base. Some seeding particles such as the core/shell NaCl-TiO$_2$ particles engineered by nanotechnology deliquesce at a relative humidity of 50% (Tai et al., 2017), while particles with low hygroscopicity deliquesce at significantly higher relative humidity (Brooks et al., 2002).

Because in the sub-saturated and near saturated environment the integration of the diffusional growth equation of aerosols is rather time-consuming in most of the numerical models, even in the relatively simple parcel model (e.g., Cooper et al, 1997), the nucleation process is often parameterized. Different techniques are available to estimate the droplet spectra just after the



activation. In most bin microphysical models, only the number of the activated CCN particles is calculated. All activated droplets are put into one bin (e.g., Rasmussen et al., 2002), or distributed over multiple bins using a prescribed function (Reisin et al., 1996). Kogan (1991) presented a method for evaluating the droplet spectra at the cloud base by using prescribed growth factors. This method was used by Yin et al. (2002) and Xue et al. (2010, 2012).

A general finding about hygroscopic seeding is that the seeding efficiency is proportional to the size of the seeding particles (Cooper et al., 1997; Yin et al., 2002; Kuba and Murakami 2010). These numerical studies found negligible effects if the radius of seeding particles is less than 1 μm, and strong effects if the size is around 10 μm. In these models, the seeding particles are assumed to be some types of salts with the hygroscopicity parameter (κ) from 0.6 to 1.2. Tessendorf et al. (2021) ran a parcel model to study the impact of the hygroscopic flare seeding in the Queensland Cloud Seeding Research

Program (QCSRP; Tessendorf et al. 2012). They found pronounced seeding effects in continental-influenced cloud conditions. Tai et al. (2017) introduced a novel hygroscopic material (NaCl-TiO$_2$) engineered and produced by nanotechnology. These particles are henceforth referred to as "nanoparticles". They consist of a shell of TiO$_2$ around a NaCl core so that the water vapor is channelled through TiO$_2$ to the NaCl, and droplets are formed in a very low relative humidity environment ($\approx$ 50%). The hygroscopicity parameter is very high at the onset of the deliquescence. Theoretically, the κ value

of such material depends on the growth factor of the particles, defined as the ratio of the particle's wet and dry radii. However, no laboratory experimental data is available to verify this dependency.

The large-scale impact of cloud seeding on the complex dynamical and microphysical system has been studied via 3D simulations and field experiments. However, the small-scale phenomena (both in time and space) cannot be easily resolved by these models. Parcel model simulations can fill the gap between the laboratory and field observations and/or 3D

simulations. The parcel models with detailed microphysical schemes reduce the uncertainty caused by microphysics-dynamics interactions. They have been proven a useful tool to study microphysical or other physical processes (e.g., Klinger et al., 2018). In this study, we focus on the warm microphysical processes near the cloud base. The Lagrangian nature of the parcel model allows for an accurate representation of the sharp change of supersaturation near the cloud base, which determines the aerosol activation and the evolution of DSD near the cloud base.

The recent field campaign conducted by the Stratton Park Engineering Company (SPEC Inc.) in the United Arab Emirates (SPEC UAE hereafter), under the UAE Research Program for Rain Enhancement Science (Mazroui and Farrah, 2017), recommended using numerical models 1) to unravel the role of coarse background particles in precipitation formation in natural clouds, and 2) to understand how the efficiency of seeding depends on the size distribution and hygroscopicity of the background aerosol particles and that of the seeding materials (Wehbe et al., 2021). To better understand the hygroscopic

seeding impacts on clouds and precipitation in different environment, the variability of the background aerosol should be considered (Bruintjes 1999; Tessendorf et al. 2012; Flossman et al. 2019). The purpose of this research is twofold:

(i) Understanding the mechanism of the spectral broadening induced by characteristics of both background aerosols and hygroscopic seeding materials. A large number of numerical experiments are performed to examine the dependence of seeding efficiency on the type of seeding materials, on the cloud dynamics, and on the





characteristics of background aerosols. We hypothesize that the competition for the available vapor near the cloud base between the drops formed on fine-mode (r < 0.1 $\mu m$) and coarse-mode particles (r > 1 $\mu m$) impacts the precipitation formation.

(ii)      Addressing the impacts of different seeding materials on DSD and exploring the efficiency of hygroscopic seeding.

The structure of the paper is as follows. Section 2 introduces the parcel model with a novel hybrid bin microphysics scheme.

The observational data used in this numerical study are described in section 3. Analysis of the evolution of the DSDs for both control and seeded cases are given in section 4. The results are summarized and the conclusions are provided in section 5.

## 2 Methods

A hybrid numerical microphysics scheme combining moving and fixed bins is used to simulate the evolution of drop

spectrum in an ascending air parcel. A bin scheme with moving boundaries is applied in the region where the diffusional growth of water drops is the dominant microphysical process. This region extends from a few hundred meters below the cloud base to the level where the air parcel reaches the altitude of 100 m above the cloud base (Fig. 1). The advantage of the moving bin method is that it inherently conserves the number concentration of particles and allows the tracking of aerosol mass inside of drops without numerical diffusion in mass and spatial dimensions. Different types of aerosol particles are

considered in the initial aerosol spectrum for a realistic representation of solute effect. Equation 1 is integrated numerically at each bin boundary, and the amount of condensed vapor was calculated by assuming linear size distribution within the bins (Geresdi and Rasmussen, 2005):

$$\frac{dm_{d,i,k}}{dt} = 2\pi D_{d,i,k} \frac{\left[ S - 1 - \frac{4\sigma_{wa}}{R_v T \rho_w D_{d,i,k}} + \kappa_k \frac{m_{ap,i,k}}{\left(m_{d,i,k} - m_{ap,i}\right)} \frac{\rho_w}{\rho_{ap,k}} \right] \cdot f_v}{\frac{L_v}{k_a^* T}\left(\frac{L_v}{R_v T} - 1\right) + \frac{R_v}{e_{sat,w} D_v^*}} \quad , \quad (1)$$

where $m_{d,i,k}$, $D_{d,i,k}$ and $m_{ap,i,k}$ are the mass and the diameter of the water drop and the mass of the aerosol particle at $i$-th bin

boundary. The index $k$ represents the type of the aerosol particles on which drops formed. The $\kappa_k$ hygroscopicity parameter depends on the chemical composition of aerosol particles of type $k$ (Petters and Kreidenweis, 2007). $\rho_w$ and $\rho_{ap,k}$ are the density of water and that of dry aerosol particles, respectively. The impact of the ventilation effect ($f_v$) and the gas kinetic effect ($k_a^*$ and $D_v^*$) is also taken into consideration. The diffusional growth was calculated by integrating Eq. 1 using a time step of $10^{-4}$ sec. The advantage of such a small time step is the elimination of the potential error that caused by

overestimating the impact of the solution and Kelvin effect. Furthermore, it allows a more realistic simulation of the competition for the available vapor among the water drops containing different chemical compounds and of different sizes. Because the diffusional growth of wet aerosol particles is calculated before the drops surpass the critical size, no arbitrary





assumption for the number concentration of the activated aerosol particles or the size of drops formed on the activated aerosol particles is needed.

At 100 m above the cloud base, the size of drops is large enough to neglect both the curvature and the solution terms, so Eq. 1 can be simplified to Eq. 2:

$$\frac{dm_{d,i}}{dt} = 2\pi D_{d,i} \frac{(S-1)\cdot f_v}{\frac{L_v}{k_a T}\left(\frac{L_v}{R_v T}-1\right)+\frac{R_v}{e_{sat,w}D_v}}  \qquad, \qquad (2)$$

where $m_{d,i,}$ and $D_{d,i,}$ are the mass and the diameter of the water drop at $i$-$th$ bin boundary, respectively.

Collision-coalescence can start to play a role in the evolution of the DSD at this stage. Therefore, the fixed bin scheme is

more appropriate, in which the size distributions of drops formed on different types of aerosol particles are merged and redistributed into fixed bins. In the case of the fixed bin scheme, growth by condensation and by collisional-coalescence is calculated as in Geresdi and Rasmussen (2005). Equation 2 is integrated using a time step of 0.1 sec. Accretion of drops due to collision-coalescence is calculated with a time step of 1.0 sec.

For the initial conditions, the size distributions of the aerosol particles in different categories are divided into 70 bins ranging

from 0.016 μm to 46.6 μm. In the case of the fixed bin scheme, 48 bins are defined over the size range from 0.1 μm to 5.0 mm.

The role of coarse particles in the precipitation formation has been studied in many researches (e.g., Jensen and Lee 2008, Witte et al., 2017). In the numerical models, it is generally assumed that coarse particles could form large drops at the cloud base and accelerate the collision-coalescence. However, field observations show that presence of the hygroscopic coarse

particles does not necessarily initiate the efficient accretion of water drops (e.g., Wehbe et al., 2021). We proposed a parameter to describe the impact of coarse particles on the evolution of the DSD using the following formula:

$$H_c = \kappa_c \frac{n_c}{n_{act}}, \qquad (3)$$

where $\kappa_c$ and $n_c$ are the hygroscopicity parameter and the number concentration of coarse particles, respectively. $n_{act}$ is the number concentration of all activated particles. This idea is based on the earlier findings that increasing the concentration of

the CCN would suppress the impact of the coarse particles: Cooper et al. (1997) and Segal et al. (2004) found that warm rain formation can be enhanced by coarse-mode seeding particles, with the seeding efficiency depending on the concentration of the activated background aerosol particles. Flossmann and Wobrock (2010) confirmed via numerical simulations that an increase of CCN concentration reduced the efficiency of precipitation formation, and the role of giant aerosol particles was less evident. However, the mechanism responsible for broadening the DSD was not clarified in these studies. We

hypothesize that diffusional growth near the cloud base is determined by the competition for the available vapor among the fine and coarse particles, which directly impacts the width of the DSD. The efficiency of the subsequent collision-coalescence depends on the resultant DSD affected by this broadening mechanism.





A typical way to evaluate seeding efficiency is to calculate the change of surface precipitation. However, it is not feasible in parcel model simulations. Generally, the DSDs from numerical experiments are compared to draw conclusions (Cooper et al., 1997; Tessendorf et al., 2021). Analyzing a large number of numerical experiments requires applying parameters that are both easy to calculate and sensitive to the broadening of the DSD. Grabowski and Wang (2009) used the reflectivity parameter (6th moment of the DSD) to describe the broadening of the tail of the DSD. In this study we follow Grabowski and Wang (2009) and use the difference in the reflectivity values 1 km above the cloud base between the control ($Z_{cntrl}$) and seeded ($Z_{seed}$) cases to measure the impact of seeding:

$$E = log\left(Z_{seed}\right) - log\left(Z_{cntrl}\right) \quad , \qquad (4)$$

## 3 Observational data

The data used in this study are obtained from two weather modification field campaigns. The first project is the aforementioned QCSRP, which was carried out in southeastern Queensland, Australia (Tessendorf et al., 2012). The second project (SPEC UAE) was carried out close to the Al Hajar Mountains in the UAE (Wehbe et al., 2021). The purpose of both research projects was to study the effect of hygroscopic seeding on the precipitation from convective clouds. The characteristics of aerosol particles, both background and seeding particles, and that of liquid drops were measured by different instruments. The wide variety of environmental conditions, different size distributions, and chemical compositions for both background aerosol particles and seeding materials related to these two projects allow for detailed sensitivity studies on the processes that impact the evolution of DSD in the updraft core of convective clouds. The data used in this study represent the typical conditions in these field projects.

The characteristics of background aerosol particles and seeding materials are summarized in Tables 1 – 4 and in Fig. 2 (see more details in Wehbe et al., 2021 and Tessendorf et al., 2021). In the case of the SPEC UAE campaign the hygroscopicity parameter of natural aerosol particles for the intermediate ($0.1\ \mu m \leq r_a < 1\ \mu m$) and coarse ($r_a \geq 1\ \mu m$) mode is available from earlier studies (Wise et al., 2007; Semeniuk et al., 2013). Unfortunately, reliable data on the hygroscopicity and concentration of the fine aerosol particles ($r_a < 0.1\ \mu m$) are not available. The size distribution of the soluble fine aerosol particles was modified to fit the observed number concentration of water drops (Fig. 2a). To address the uncertainty in the hygroscopicity of the aerosols, sensitivity tests on the hygroscopicity parameter were conducted (Table 1). In SPEC UAE cases, the aerosol particles in the coarse and intermediate modes were supposed to be more hygroscopic than the particles in the fine mode. The concentration of the coarse particles was about 1.0 cm$^{-3}$ (Wehbe et al., 2021). In the QCSRP, the background aerosol particles were divided into two categories, continental and maritime influenced ones, which differ in the size distribution and hygroscopicity (Fig. 2a). In the continental air mass the concentration of coarse particles was small (0.5 cm$^{-3}$) and less soluble; in the maritime air mass the concentration of coarse particles was significantly larger (5.0 cm$^{-3}$) and their hygroscopicity was higher as well.


Tables 2 – 4 and Figs. 2b, c summarize the characteristics of three different types of seeding material (nanoparticles, ICE70 and NCM) used in these field projects. The different concentrations of seeding particles under different case names in these tables reflect the dilution effect due to dispersion and diffusion of seeding particles from the releasing source up to the cloud base. Laboratory observations show that the nanoparticles start to deliquesce at a low relative humidity due to their large hygroscopicity (Tai et al., 2017). However, the hygroscopicity of this material is not constant and decreases rapidly as the

outer $TiO_2$ shell dissolves and the growth factor increases. Unfortunately, quantification of this hygroscopicity-vapor relationship is unknown. To take this uncertainty into consideration, the following formula is used to represent the dependence of the hygroscopicity parameter on the size of drops formed on the nanoparticles:

$$\kappa = \kappa_1 + (\kappa_0 - \kappa_1) exp(-X \cdot (GF - 1)) \quad , \qquad (5)$$

where $\kappa_0 = 20.0$ is the hygroscopicity of dry nanoparticles and $\kappa_1 = 1.12$ is the hygroscopicity of salt particles, and $GF$ is the

growth factor. Different $X$ values were tested to examine the sensitivity to the decreasing rate (Table 2). When $X = \infty$, the formula represents salt particles while $X = 0$ indicates ideal nanoparticles.

Table 3 lists four cases of the hygroscopic flares of ICE70. We include dilution factors of 10 and 100 to test the impact of dispersion and diffusion. We adopt two size distributions taken from the above two campaigns to include the measurement uncertainty of this seeding material (see also Fig. 2c). The discrepancies between the size distributions are the consequence

of the different observation techniques applied in the two field projects (Bruintjes et al., 2012; Wehbe et al., 2021).

The NCM seeding material (Table 4 and Figure 2c) is released from a recently developed hygroscopic flare. It was applied in the UAE field program. The number concentrations are given at dilution factors of 10 and 100.

Figure 1 shows the temperature and vertical velocity profiles of the ascending air parcel for the two ICE70 cases with different size distributions (ICE70_2 and ICE70_4). The temperature changes dry and wet adiabatically under and above the

cloud base, respectively. Different updraft vertical profiles (red curves in Fig. 1) were generated to represent the possible cloud dynamics suitable for convective clouds. From bottom to top, the horizontal lines show 1) the level of RH = 70%, where the calculation of the diffusional growth starts, 2) the level of the cloud base (blue lines), 3) the level at 100 m above the cloud base, and 4) the level at 1 km above the cloud base, respectively. The updraft profile denoted by $w_1$ in Fig. 1a was adapted from a high-resolution WRF simulation for the SPEC UAE case (Lee et al., 2021). $w_2$ is half of $w_1$. For QCSRP

cases in Fig 1b, constant vertical profiles with values of 1, 2, and 5 m s$^{-1}$ are used in this study.

## 4 Results

### 4.1 Control cases

Using nine different background aerosol conditions (Table 1) and three different updraft profiles for both field programs (Fig. 1), 27 control cases are simulated. Figure 3a shows the relation between the $H_c$ parameter (Eq. 3) and reflectivity at 100

m above the cloud base. This panel reveals how coarse particles impact the drop size distribution when only the diffusional



growth is taken into consideration. In real cases, the diffusional growth can be considered dominant under this level; the role of the collision–coalescence is negligible. The monotonic increase of the reflectivity with increasing $H_c$ shows that broadening of the size distribution not only depends on the number concentration of the coarse particles but is also affected by the hygroscopicity of the coarse particles. Furthermore, the formation of big drops on coarse particles is constrained if the fraction of coarse particle is small compared to the number concentration of the activated aerosol particles.


Figure 3a also reveals that the reflectivity is negatively correlated with the updraft velocity near the cloud base (see the symbols with different forms and colors). To avoid overcrowding, only three cases (BGUAE_2, BGQNC_1 and BGQNM_1) are highlighted by using different symbols in Fig. 3. The relation between the reflectivity and updraft velocity is similar in other cases. This implies that the larger updraft velocity may suppress the broadening of the DSD tail. Tessendorf et al.


(2021) found a similar correlation between the updraft velocity and the DSD broadening near cloud base. They argued that the broadening is the consequence of a longer time period of condensation growth in a weaker updraft (i.e., residence time effect). Our results suggest that the broader DSD is instead dominated by the Ostwald-ripening effect (e.g., Yang et al., 2018).

A numerical experiment is conducted to demonstrate this. In this experiment, the initial concentration of aerosol particles is


the same as in the BGUAE_2 case. Collision–coalescence was not considered in this calculation, and the moving bin scheme was used to calculate diffusional growth of the drops (Eq. 1) during the whole simulations up to 1 km above the cloud base. The temperature profile and vertical velocity profiles of $w_1$ and $w_3$ (with cloud base updraft speed of about 5 and 1 m s$^{-1}$, respectively) in Fig. 1a are used for the simulation. Figure 4a shows the DSDs at 100 m and 1 km above the cloud base. At 100 m above the cloud base the DSD under the weaker updraft is significantly wider, but this difference decreases at 1 km


above the cloud base, which demonstrates that the residence time has no significant effect on the evolution of the DSD if a deeper region is considered. Tessendorf et al. (2021) studied the evolution of DSD within only 400 m above the cloud base. Figure 4c shows the evolution of drop sizes with different initial dry radii. In weak updraft (vertical velocity profile denoted by $w_3$), coarse particles become significantly larger, and the growth of fine particles is less affected by the change of updraft velocity. In addition, the aerosol particle with initial radius of 0.05 μm becomes slightly smaller if the updraft is weaker.


The profiles of the liquid water content (LWC) and supersaturation in Figure 4b further support our argument. In weak updrafts, the LWC becomes equal to the adiabatic LWC immediately above the cloud base. In the strong updraft (updraft profile of $w_1$) the vapor flux exceeds the vapor depletion due to the diffusional growth. Therefore, LWC reaches the adiabatic value only at a higher elevation, at about 100 m above the cloud base. The actual LWC at 50 m above the cloud base was about 15% smaller than the adiabatic value. This deviation from the adiabatic liquid water content can be explained


by the shorter residence time as asserted by Tessendorf et al. (2021). The stronger updraft yields a larger supersaturation, which reduces the role of both the curvature and solution effects on the diffusional growth. In contrast, in the case of weak updraft, due to the smaller supersaturation, the diffusional growth of drops formed on larger and/or more soluble particles can grow faster against the drops formed on the fine particles (Ostwald-ripening effect).



The plots in Figure 4d demonstrate the ripening effect. The vertical regions where the ripening works can be assigned by the
condition where drops formed on larger and/or more hygroscopic aerosol particles grow faster than drops formed on the smaller and/or less hygroscopic aerosol particles. In the case of the weak updraft ($w_3$) drops formed on the coarse particles growth faster than the drops formed on the fine particles in the whole plotted range both under and above the cloud base except for a narrow interval above the cloud base. If the updraft is stronger (solid lines in Figure 4d) the drops formed on the coarse particles grow faster only under the cloud base, above the cloud base only the drops formed on giant CCN ($r_0 = 5$ μm)
grow faster than the drops formed on the smaller particles. At about 100 m above the cloud base the drops are large enough to neglect both the solution and curvature effects, and the rate of diffusional growth is proportional to supersaturation. Because in the adiabatically ascending air parcel, the supersaturation is positively correlated with the updraft velocity (e.g., Rogers and Yau, 1996), the longer residence time is balanced by the smaller supersaturation.

To examine the effect of coarse particles on DSD broadening, we investigate the relation between $H_c$ and reflectivity at 1 km
above cloud base (Fig. 3b). Collision-coalescence is considered after 100 m above the cloud base, as the drops are large enough to initiate the process. The relation is not as straightforward as in Fig. 3a. In the strong updraft, the reflectivity is between -20 and -10 dBZ, and they do not depend on the value of $H_c$. It means that in the case of intense updraft, the droplet size distributions only slightly depend on the presence of the coarse particles even if the impacts of collision-coalescence are also taken into consideration. The tail effect of coarse particles is weak, and the DSD broadening depends only on the
number concentration of activated aerosol particles. This is found even in the maritime case (BGQNM_1), where a relatively high concentration of hygroscopic coarse particles (N=5 cm$^{-3}$, $H_c \approx 1.5 \times 10^{-3}$) is present. This result agrees with the finding in SPEC UAE ($H_c$ is between $3 \times 10^{-4}$ and $7 \times 10^{-4}$). They found that DSDs remained narrow, although the concentration of the coarse particles was around 1 cm$^{-3}$ (Wehbe et al., 2021). The sensitivity to the presence of coarse particles was found to be important only if the updraft was weak ($w_3$ in Fig. 1). The broader DSD formed due to the Ostwald-ripening effect
promotes the collision-coalescence process. The fact that even in the case of weak updrafts the DSD remains narrow if the value of $H_c$ is small suggests that the diffusional growth remains dominant, and the impact of collision-coalescence is small or negligible. This scenario is represented by the BGUAE_4 case (there are no coarse particles in the aerosol spectrum). The reflectivity 1 km above the cloud base remains small ($\approx$ -16.0 dBZ), independently of the updraft profile.

### 4.2 Effect of seeding

In this section, the results of the impact of different seeding materials on the evolution of DSD near the cloud base are presented. Because the characteristics of seeding materials of ICE70 and NCM are rather similar (see Tables 3 – 4, furthermore Fig. 2c), the results about these materials will be discussed together.

In the seeded cases the impact of coarse particles is described by the modified version of Eq. 3:

$$H_{c,seed} = \frac{\kappa_{c,bg} \cdot n_{c,bg} + \kappa_{c,seed} \cdot n_{c,seed}}{n_{act,bg} + n_{act,seed}} \qquad , \qquad (6)$$



where $\kappa$ and $n$ are the hygroscopicity parameter and the number concentration of coarse particles. $n_{act}$ refers to the number concentration of activated particles. The subscripts of $bg$ and $seed$ indicate the background aerosol and seeding materials, respectively.

### 4.2.1 Effect of nanoparticles

The nanoparticles have a core-shell structure of NaCl-TiO$_2$, which can adsorb water vapor and deliquesce at a lower
environmental RH than the pure NaCl (Tai et al., 2017). The size of the nanoparticles is comparable with that of background coarse particles. Fig. 5 a – c shows how seeding nanoparticles impacts the DSD at 1 km above the cloud base in three different background aerosol conditions. Seeding results in a broader DSD in all cases, independent of the background aerosol size distributions applied in this study. However, the broadening effect is found to be negatively associated with the concentration of the background coarse particles. The increase of the concentration of drizzle size drops (r > 25 µm) is
highest in the seeded BGQNC_1 case and lowest in the seeded BGQNM_1 case (Fig. 5b, c). In these cases, the concentration of the background coarse particles is 0.5 and 5.0 cm$^{-3}$, respectively. The seeding has no significant effect on the growth rate of the drops formed on background aerosol particles either in the seeded BGUAE_1 or BGQNM_1 (Fig. 6a, c). However, the growth surplus for the nano particles due to the ripening effect is significant in the seeded BGUAE_1 case (Fig. 6a, b). In the seeded BGQNM_1 case, the growth rate of the nano particles reduces by half compared to the seeded BGUAE_1 case,
and the growth rate surplus is significantly smaller compared to the growth rate of background aerosol particles (Fig. 6c, d). Note that the neglect of the curvature and solution effect for the natural giant CCN and the nano particles beyond 100 m above the cloud base may results in an underestimation of the Ostwald-ripening effect (Fig. 6b, c).

We also examined the impact of the hygroscopicity of the nano seeding material by changing the value of $X$ value in Eq. 5. It is found that the difference between NANO_1 and NANO_5 (hygroscopicity decreases quickly with the increasing wet size)
is small (Fig. 5a – c). The impact of NANO_2 (hygroscopicity decreases slowly with the wet size) is substantial even in the BGQNM_1 case (Fig. 5a – c). Note, using X = 0.1 in NANO_2 may overestimate the hygroscopicity of the real nanoparticles. The test also shows that if the concentration of the seeding particles was very small (0.12 cm$^{-3}$ in NANO_4), seeding effect was significant both in BGUAE and BGQNC cases, but it was small in BGQNM cases. The reflectivity values of seeded/control cases with updraft profile of w$_1$ at 1000 m above the cloud base are as follows: -1.37/-16.42, -9.43/-19.88,
and -15.47/-17.56 dBZ in the BGUAE_1, BGQNC_1 and BGQNM_1, respectively. This finding suggests that the concentration of the nanoparticles should be comparable or larger than that of the background coarse particles to have a significantly positive seeding effect, in agreement with the recommendations in Wehbe et al. (2021) based on the aircraft observations.

The seeding by nano particles can have significant seeding effect in all cases, and the effect becomes stronger as the updraft
is weaker. This agrees with the results in the previous section, because the injection of the nanoparticles increases the concentration of highly hygroscopic coarse particles. The variation in seeding efficiency due to change in $X$ (Eq. 5) significantly decreases as the updraft decreases. For example, with the aerosol size distribution of BGUAE_1, the efficiency





of seeding (Eq. 4) increases from 18.8 to 35.1 dBZ as the $X$ changed from infinity (salt particle) to zero (ideal nanoparticle) for the case with an updraft around 5 m s$^{-1}$ at the cloud base. This efficiency changes to 34.1 ($X = \infty$) and 41.5 dBZ ($X = 0$)

when the updraft at the cloud base was about 2 m s$^{-1}$. This means that in the case of strong convective clouds the nanoparticles with large hygroscopicity can be more favorable than the pure salt particles, and in weaker updrafts this benefit decreases significantly.

### 4.2.2 Effect of flare particles (ICE70 and NCM)

Figures 5 d – f show how the ICE70_4 reagent affects the DSD at 1 km above the cloud base. The impact of NCM, with a

similar seeding particle concentration to ICE70, is also similar. The differences in the seeding efficiency between the two seeding materials are less than 1 dBZ. In the plotted cases the updraft velocity is 1 m s$^{-1}$ (w$_3$ profiles in Figure 1a and b), and the dilution factor is 100. For a stronger updraft or a dilution factor of 10 (i.e., a higher concentration), the seeding effect is smaller or even negative.

It is also found that the seeding efficiency of the flare particles is significantly smaller than that of the nanoparticles. The

difference between the hygroscopicity cannot explains this difference, because even the effect of salt particles, whose hygroscopicity is nearly the same as that of the flare particles, is positive (see Figs. 5 a-c). However, the average size of these two types of seeding materials was different (Fig. 2). Fig. 7a shows how the equilibrium size of water drops depends on $\kappa$ and the size of the aerosol particles. The drop size versus hygroscopicity curves at 1 and 5 seconds after the diffusional growth are also plotted. While the presence of aerosol particles with a size of 0.1 μm cannot contribute to the drop formation

larger than 5 μm at near the cloud base, the drops formed on the aerosol particles with a size of 2.5 μm can reach 10 μm within 1 second if $\kappa$ is near 1. The equilibrium size is almost independent of $\kappa$ if the aerosol size is submicron, and it is more sensitive to the hygroscopicity in the case of the coarse particles. The importance of the size of seeding materials over the hygroscopicity is shown in Fig. 7b. The background aerosol concentration is BGUAE_3 (Table 1). The seeded cases with NANO_7 (with extremely high hygroscopicity but a very small mean radius) and NANO_8 (with a smaller hygroscopicity

but a large mean radius) are compared (Table 2). Although the hygroscopicity in the NANO_7 case is extremely high, the seeding effect is negligible. When the size of the seeding material is increased, despite that the hygroscopicity is significantly reduced, the seeding effect is evidently larger. The weaker seeding effect by flares particles is also well demonstrated in Figs. 6 e and f. While flare particles suppress the growth of the background coarse particles (less extent) and aerosol particles with size of 0.1 μm (larger extent), their growth rate surplus comparing to the background particles due to the ripening effect

is negligible.

### 5 Discussions and conclusions

In this study, the impact of background aerosol and seeding particles on the evolution of DSD was studied by using a parcel model with a novel hybrid bin microphysics scheme. The applied moving bin approach is able to avoid the numerical





diffusion problems that an Eulerian bin scheme produced (Grabowski et al. 2019; Morrison et al. 2020). A large number of

numerical simulations were performed to investigate how hygroscopic seeding can affect the DSD of convective clouds in various environmental conditions. The results are valid only if the fall out of the hydrometeors formed in the ascending air parcel is negligible, so the sensitivity test was not extended to updraft speeds less than 1 m s⁻¹.

To describe the impact of the coarse particles on the evolution of the drop spectrum, a new parameter dependent on both the hygroscopicity of coarse particles and their relative concentration to that of all activated aerosol particles was introduced.

Evidence from observation and numerical model simulation indicates that the presence of large, highly soluble aerosol particles (coarse particles) can form big drops and initiate the collision-coalescence process efficiently (e.g. Witte et al., 2020), i.e., the tail effect. Our results about the control cases show that the impact of the coarse particle on the evolution of the DSD depends on the updraft velocity. When the updraft is about 1 m s⁻¹, the early broadening of the drop size distribution is more sensitive to the presence of the coarse particles. In this case the smaller supersaturation and the presence

of coarse particles (depending on their hygroscopicity) enhance the Ostwald-ripening effect, which can significantly impact the evolution of the DSD near the cloud base. When the updraft at the cloud base is strong (≈ 5 m s⁻¹), background coarse particles do not play important role in broadening of the DSD even with a relatively high concentration (≈ 5 cm⁻³). In the case of strong updraft (w ≈ 5 m s⁻¹), the supersaturation is large which reduces impact of the Ostwald-ripening effect and leads to a low sensitivity of DSD to the coarse particle concentration.

Figure 8a shows how the change of the $H_c$ parameter ($H_{c,seed}$ (Eq.6) minus $H_c$ (Eq.3) ) impacts the water vapor uptake near the cloud base. The vertical scale gives the ratio of LWC between the seeded and control cases at 100 m above the cloud base. The strong correlation reveals that the enhancement of the $H_c$ parameter due to the injection of highly soluble coarse particles (in this case, the number concentration of activated particles only slightly increased) can significantly increase the water vapor uptake under the cloud base (see the intense growth rate of the drops formed on nanoparticles in Figure 6b and

d). If the seeding materials contain a large concentration of fine particles, the water vapor uptake reduces. In this case, the number concentration of coarse particles does not change, and the number concentration of the activated particles remains the same or increases. Our results do not support the frequently cited hypothesis that the hygroscopic seeding significantly reduces drop formation on natural CCN (e.g., Segal et al., 2007). It was found that the increase of the CCN concentration due to the seeding was never compensated by the decrease of activation of natural aerosol particles. Notable decrease of the

background CCN concentration was only found in the case of a weak updraft (w = 1.0 m s⁻¹).

The broadening of the size distribution near the cloud base affects the subsequent evolutions of the DSDs. Figure 8 reveals that the larger the increase of the reflectivity due to the seeding at 100 m above the cloud base, the larger the increase of it at 1000 m above the cloud base. Broadening of the spectra due to the ripening effect promotes the collision–coalescence.

In a numerical study on the AgI seeding in winter orographic clouds by Geresdi et al. (2017), it was concluded that the

efficiency of seeding is negatively correlated with the efficiency of precipitation formation in the control case. Furthermore, the chance of the overseeding is not negligible. Figure 9 shows this relation in the studied hygroscopic seeding cases. Similar to the previously mentioned AgI seeding cases, the largest positive effect occurs if the reflectivity (1 km above the cloud





base) in the related control case is small. However, a small negative effect can occur even in the case of a low reflectivity in the control case, if the concentration of the seeding particles was significantly larger than the concentration of the
background aerosol particles. If the efficiency of the precipitation formation in the control case was large (BGQNM_1, $Z \approx 20$ dBZ), the seeding effect was small even if nanoparticles were injected. However, in this case, no negative seeding effect was found even if the concentration of the seeding materials was large. Reflectivity values around zero occurred in BGUAE_2 and BGUAE_3 control cases if the updraft velocity was 2 or 1 m s$^{-1}$. In these cases, the impact of the background coarse particles was weaker than in the case of the maritime influenced air mass (BGQNM_1) but larger than in other cases.
We found the largest negative seeding effect in these conditions. Seeding nanoparticles resulted in a positive seeding effect, and seeding flare particles, independently of the dilution factor, resulted in a narrower DSD than in the control cases.

The box and whisker plots in Fig. 10 show the range of seeding efficiency (Eq. 4) in the case of nanoparticles with X = 1.0 (this value seems to be reasonable for real nanoparticles), X = infinity (this means salt particles), and ICE70 flares particles. Results show that the nanoparticles have no negative seeding effect, regardless of X values. If their concentration is large
enough, they can result in a positive seeding effect even in the case of the maritime influenced air mass. The impact of flare particles (ICE70 and NCM) is more ambiguous. The range of the seeding efficiency is from -13 to +3 dBZ. The large negative values occur if the concentration of the flare particles was significantly larger than the concentration of the background aerosols. A small positive effect was found in BGQNC cases if the updraft was moderate or small ($w_2$ and $w_3$ profile in Figure 1b) at the cloud base. Because the median of the seeding efficiency is close to zero, and the difference
between the first and third quartiles is small, the impact of this type of seeding material was found to be insignificant in most of the cases.

In this research, it was found that broadening of the size distribution due to the Ostwald-ripening effect can have a significant effect on the evolution of the DSD. However, this effect can only be demonstrated if the vertical resolution resolves the sharp change of the supersaturation near cloud base (Fig. 4b). This high resolution (about 10 m grid spacing or
finer) is quite difficult to accomplish in an Eulerian model. A parameterization of this effect is to be developed in the next phase of this research. Furthermore, the implementation of the current microphysical scheme into the Lagrangian particle framework allows us to have a more realistic simulation of the early phase of precipitation formation by taking into consideration of Ostwald-ripening effect in the full cloud volume and fallout of the hydrometeors. The other deficiency of the current simulation is that the model does not involve entrainment and mixing. These processes may have a significant
effect on the evolution of the DSD (e.g., Lasher-Trapp et al. 2005; Tölle and Krueger, 2014; Grabowski and Abade, 2017; Abade et al. 2018). The implementation of these processes may also contribute to better understanding of the broadening mechanism of DSD near the cloud base.



**Code/Data availability**

The aircraft observations are archived at the UAE National Center of Meteorology. Readers can request the dataset by
contacting research@ncms.ae. The model codes and results are available upon request.

**Author contribution**

IG, LX, SC and YW conceptualized the study. YW, RB and ST supported in data collection and analysis. JL and RM helped
provide the model initial conditions. IG and NS performed the model simulations and analysis. IG, LX and SC wrote the
manuscript. All co-authors were involved in the manuscript editing and discussion.

**Competing interests**

The authors declare that they have no conflict of interest.

**Acknowledgement**

This work was supported by the National Center of Meteorology, Abu Dhabi, UAE (UAE Research Program for Rain
Enhancement Science). This material is based upon work supported by the National Center for Atmospheric Research, which
is a major facility sponsored by the National Science Foundation under Cooperative Agreement No. 1852977. The
contribution to this research by I. Geresdi and N. Sarkadi were also supported by Hungarian Scientific Research Fund
(Development and application of novel numerical model to investigate the precipitation formation in mixed phase clouds).

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



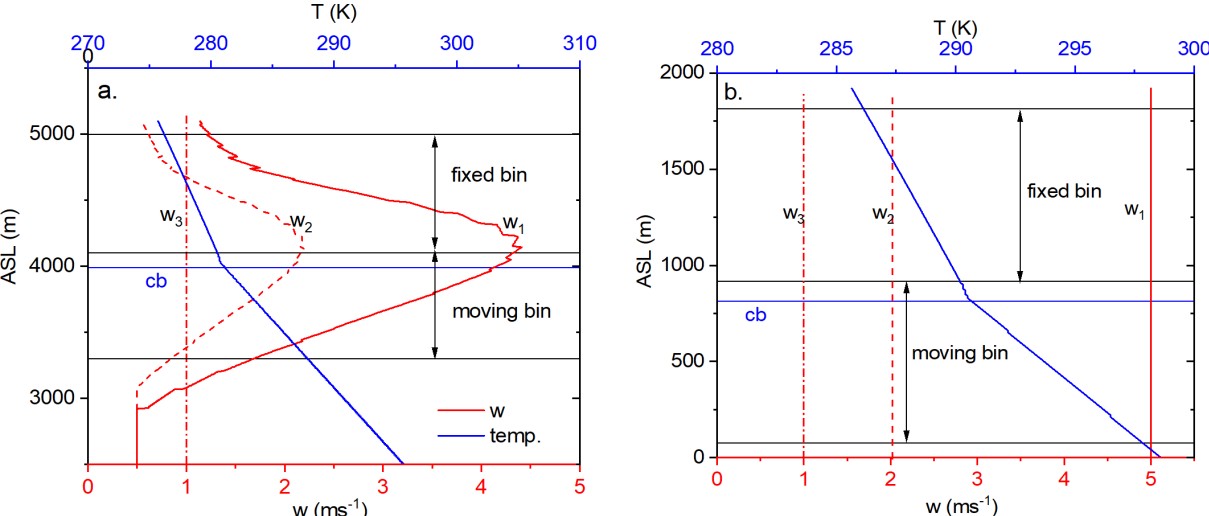

**Figure 1: The temperature profiles and the prescribed updraft profiles in the numerical simulation for (a) the SPEC UAE cases and (b) the QCSRP cases. The horizontal blue lines denote the altitude of the cloud base. The three horizontal black lines from bottom to top denote the level where the calculation of the diffusional growth starts (RH = 70%), the levels 100 m and 1000 m above the cloud base.**

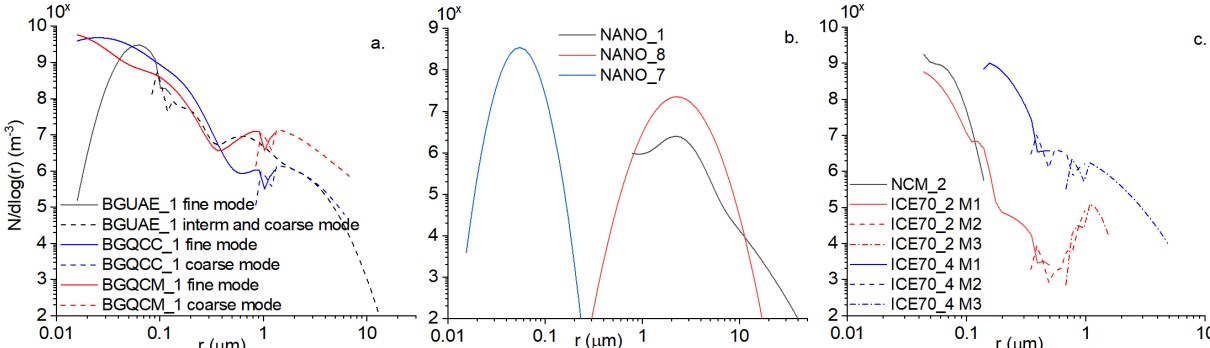

**Figure 2: Initial size distributions of (a) the background aerosols, (b) seeding nanoparticles, and (c) seeding flares. The parameters of the size distributions are given in the Tables 1–4.**





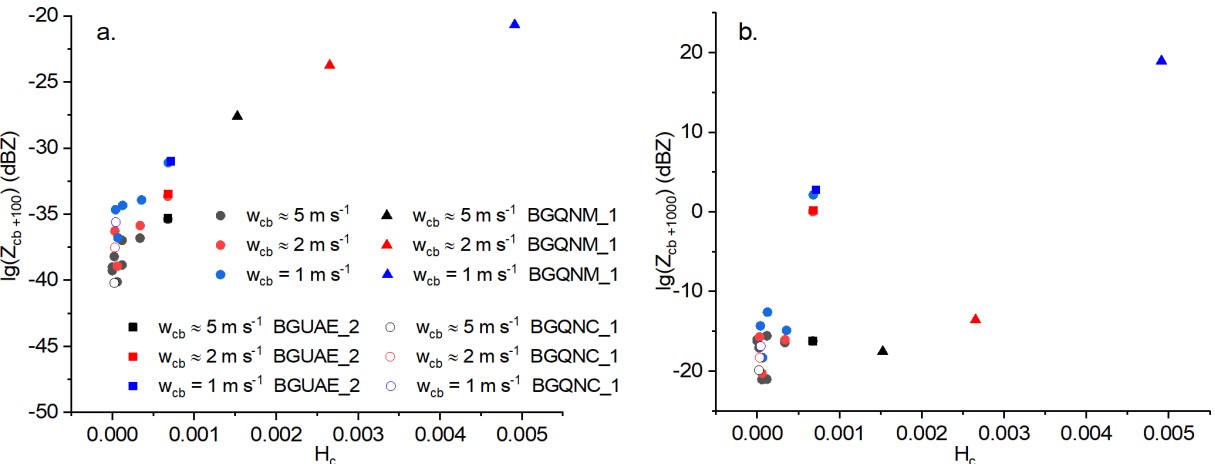

**Figure 3: Reflectivity calculated from the modelled DSD (a) 100 m and (b) 1000 m above the cloud base for non-seeded control cases. Symbols with different colours represent different updraft velocities at the cloud base as indicated by the key. Sensitivity on the updraft velocity in the case three different background aerosol size distributions is presented by different symbols (see the legend in panel a.).**






**Figure 4:** Evolution of (a) DSD, (b) vertical profiles of liquid water content and supersaturation, (c) vertical profiles of the drop radius and (d) vertical profiles of the vertical gradients of the drop radius. In panels c. and d., the lines with different colours denote different initial aerosol radius and hygroscopicity (see legend in panel c.). The solid and dashed lines denote the cases related to the updraft profiles of strong ($w_1$) and weak updrafts ($w_3$), respectively. The horizontal lines in panel d. denote the location of the cloud base (black solid line), and that of the maximum of the supersaturation in the case of updraft profiles $w_1$ (solid blue line) and $w_3$ (dashed blue line).





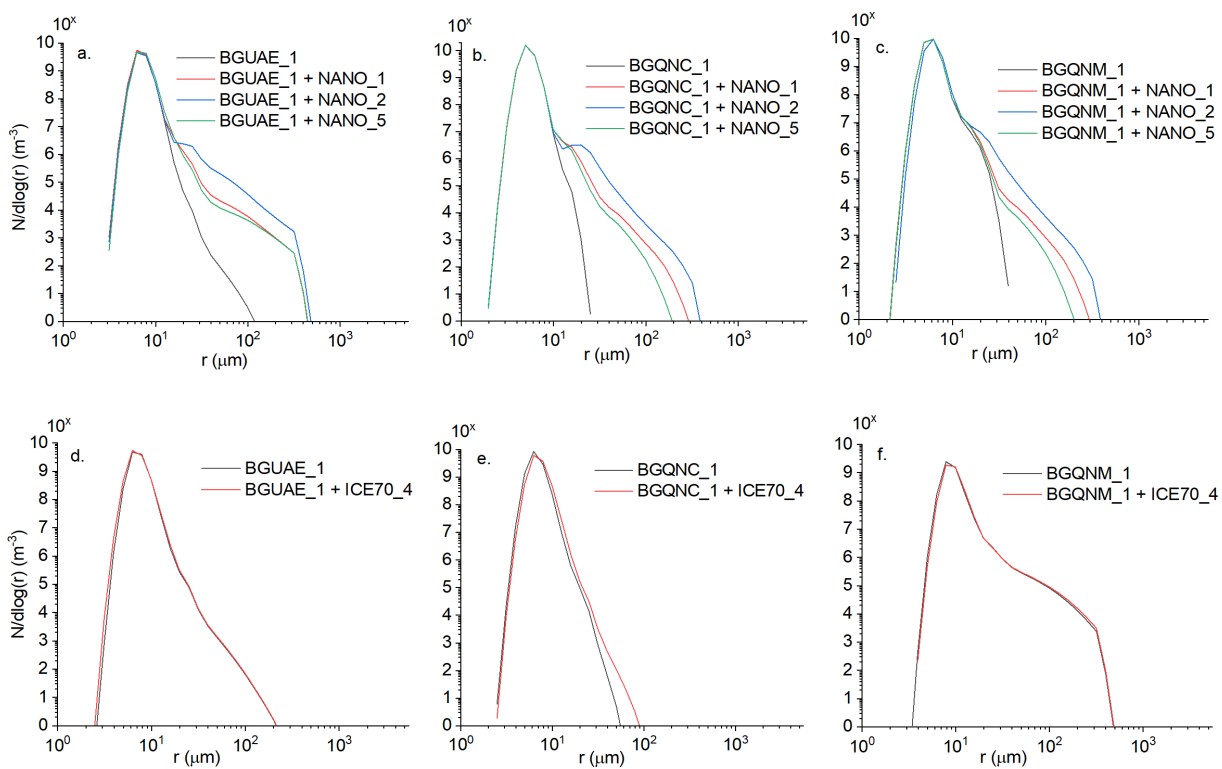


**Figure 5: Impact of hygroscopic seeding at different background aerosol size distributions and in the case different seeding materials. The black lines denote the DSD for the control cases 1 km above the cloud base. The colour lines denote the DSD formed due to the different seeding materials. The impact of the seeding of the nano particles (a., b, and c) is presented when the updraft profiles $w_1$ was used. In the case of ICE70 seeding material the DSDs are calculated at updraft profile $w_3$.**







**Figure 6:** The vertical profile of the growth rate of water drops formed on background aerosol particles with different sizes and hygroscopicity (first column) and that of drops formed on different type of seeding materials with different sizes and hygroscopicity (second column). In the left panels the solid lines denote control cases, and the dashed lines denote seeded cases. Blue and black vertical lines denote the location of the cloud base and that of the maximum supersaturation.

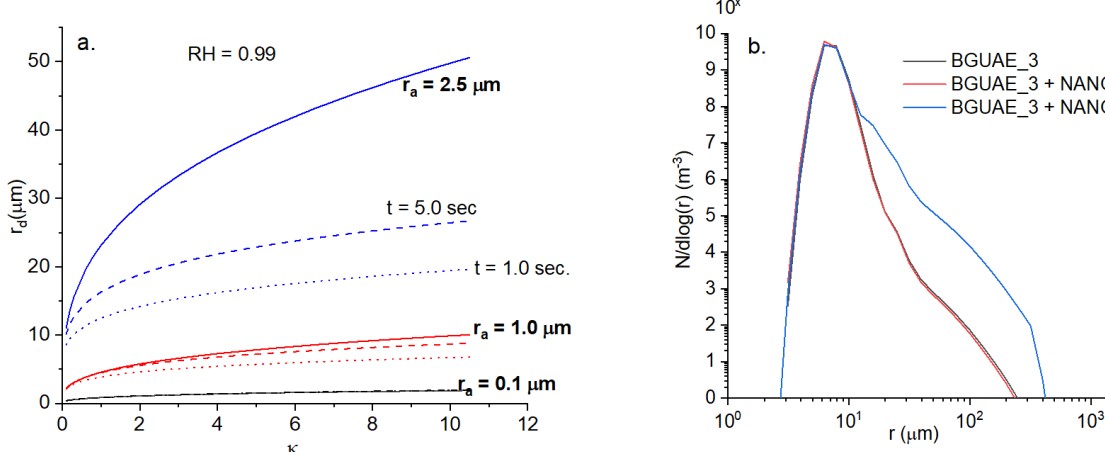

**Figure 7:** The impact of the size ($r_a$) and hygroscopicity ($\kappa$) of the seeding material. Panel a. shows the equilibrium drop sizes formed on the aerosol particles with different radius and different hygroscopicity (solid lines). Dotted and dashed lines denote the drops radius calculated by Eq. 1 after 1 and 5 sec, respectively. Panel b. shows how the small particles with large hygroscopicity (NANO_7) and large particles with smaller hygroscopicity (NANO_8) impact the evolution of the DSD. The black line denotes the DSD for the control case 1 km above the cloud base.

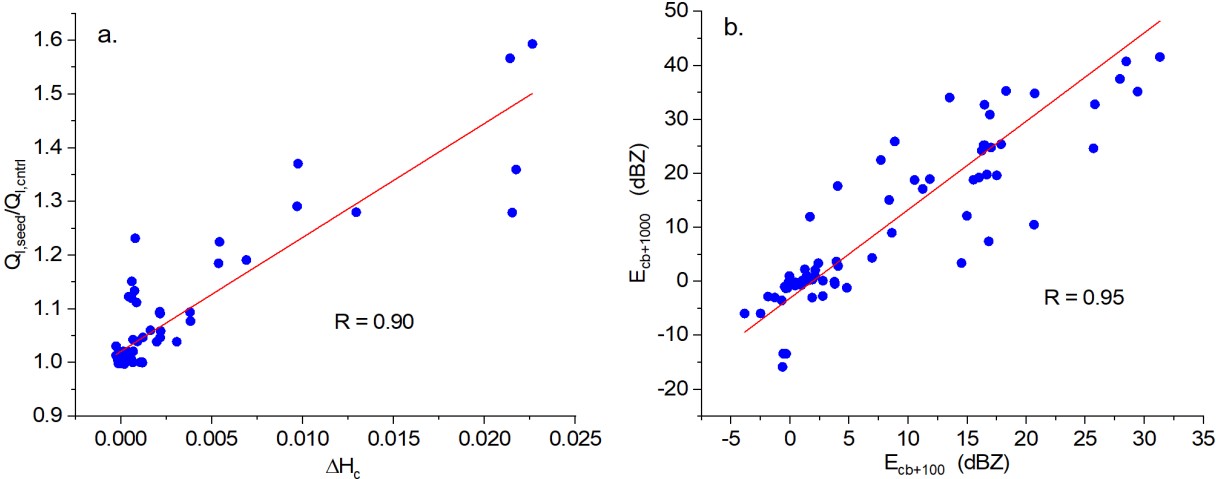

**Figure 8:** (a) The impact of the change of the $H_c$ parameter on the vapor uptake. The vertical coordinate represents the ratio of liquid water contents belong to the seeded and related control cases. (b) This panel shows how the broader size distribution generated by the Ostwald-ripening effect ($E_{cb+100}$) can result in broadening of the size distribution comparing to the control case 1000 m above the cloud base ($E_{cb+1000}$).





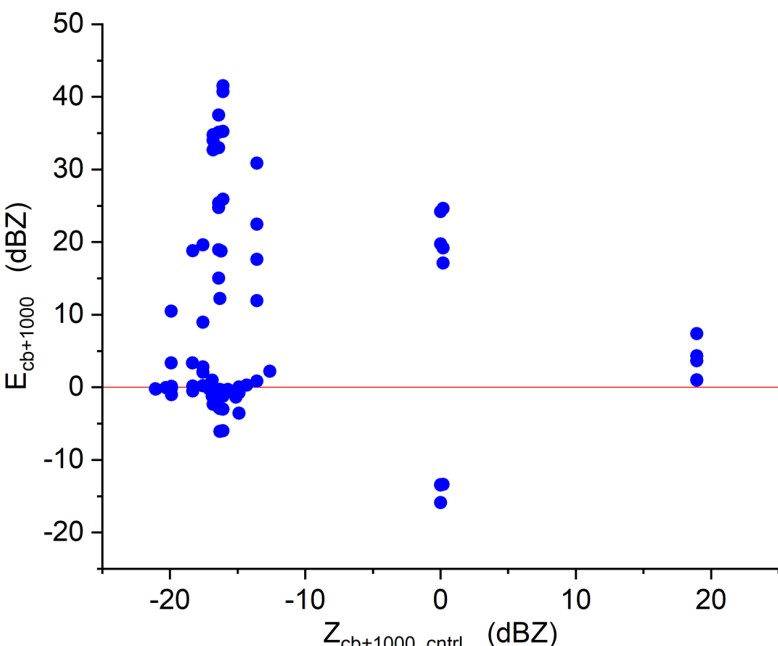

**Figure 9: The relation between the seeding efficiency ($E_{cb+1000}$) and the reflectivities for the control cases 1 km above the cloud base.**

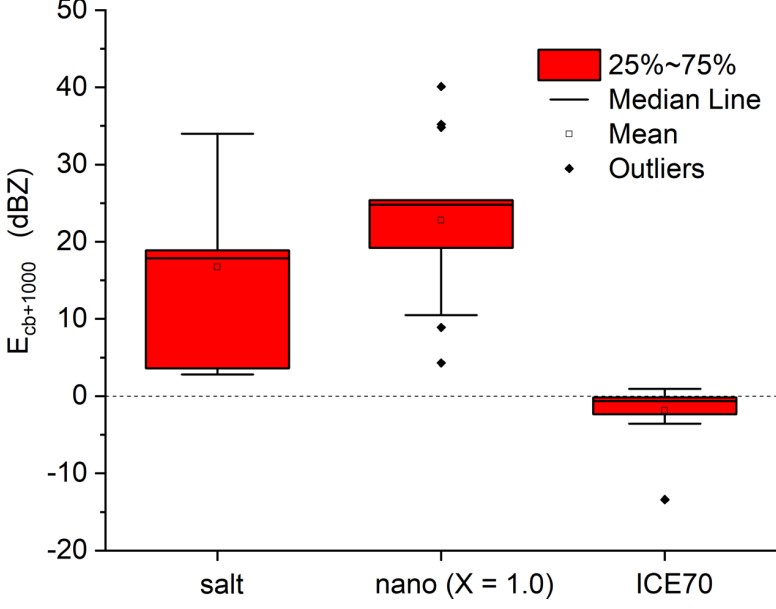

**Figure 10: The box and whisker plots presenting the efficiency range of three different seeding materials. The box plots are evaluated from data representing different background aerosol types and different updraft velocities.**

**Table 1.** Number concentration (cm⁻³) and chemical composition of the background aerosol particles observed in SPEC UAE and QCSRP. The hygroscopicity parameters for intermediate and coarse modes in the BGUAE cases were estimated





value based on the chemical composition observed by Semeniuk et al. (2014). The number concentrations are taken from the SF4 case in Wehbe et al. (2021). BGUAE means background aerosol observed in SPEC UAE campaign. BGQNC and BGQNM mean background aerosol in continental and maritime influenced air mass, respectively in QCSRP. * indicates that the observed size distribution are modified for the sensitivity test.

| Name | Number concentration of fine, intermediate and/or coarse mode (cm$^{-3}$) | Chemical composition | κ of fine, intermediate and/or coarse mode | r range (µm) |
|---|---|---|---|---|
| BGUAE_1 | 950 | internally mixed MCS | 0.1 | 0.01 – 0.10 |
| | 70 | salt coated mineral | 0.3 | 0.10 – 1.00 |
| | 1.2 | salt coated mineral | 0.3 | 1.00 – 12.1 |
| BGUAE_2 | 950 | internally mixed MCS | 0.1 | 0.01 – 0.10 |
| | 70 | salt coated mineral | 0.6 | 0.10 – 12.10 |
| | 1.2 | salt coated mineral | 0.6 | 1.00 – 12.1 |
| BGUAE_3 | 950 | internally mixed MCS | 0.3 | 0.01 – 0.10 |
| | 70 | salt coated mineral | 0.6 | 0.10 – 1.00 |
| | 1.2 | salt coated mineral | 0.6 | 1.00 – 12.1 |
| BGUAE_4* | 950 | internally mixed MCS | 0.1 | 0.01 – 0.10 |
| BGUAE_5* | 2850 | internally mixed MCS | 0.1 | 0.01 – 0.10 |
| | 84 | salt coated mineral | 0.3 | 0.10 – 1.00 |
| | 0.6 | salt coated mineral | 0.3 | 1.00 – 12.1 |
| BGUAE_6* | 2850 | internally mixed MCS | 0.3 | 0.01 – 0.10 |
| | 84 | salt coated mineral | 0.6 | 0.10 – 1.00 |
| | 0.6 | salt coated mineral | 0.6 | 1.00 – 12.1 |
| BGQNC_1 | 2930 | sulfur bearing | 0.3 | 0.01 – 1.00 |
| | 0.5 | mineral dust | 0.1 | 1.00 – 7.9 |
| BGQNM_1 | 1800 | sulfur and salt bearing | 0.4 | 0.01 – 1.00 |
| | 5.0 | sea salt and mineral dust mix | 0.5 | 1.00 – 7.9 |
| BGQNC_2* | 1410 | sulfur bearing | 0.3 | 0.01 – 1.00 |
| | 0.3 | mineral dust | 0.1 | 1.00 – 7.9 |

**Table 2.** Characteristics of the nano particles. Because the shape of the size distribution function is the same in the NANO_1 – NANO_6 cases, only the NANO_1 is plotted in Figure 2b. The mode radius is 2.0 µm in these cases. The meaning of $X$ can be found in Eq. 5. NANO_7 and NANO_8 are hypothetical particles for sensitivity study with the mode radius of 0.05 µm and 2.5 µm, respectively (blue and red curves in Figure 2b).

| Name | Number concentration (cm$^{-3}$) | Chemical composition | Hygroscopicity (κ) | X | r range (µm) |
|---|---|---|---|---|---|
| | | | | | |





| | | | | | |
|---|---|---|---|---|---|
| NANO_1 | 1.2 | NaCl-TiO$_2$ | 1.12 - 20 | 1.0 | 0.75 – 12.00 |
| NANO_2 | 1.2 | | | 0.1 | |
| NANO_3 | 12.0 | | | 1.0 | |
| NANO_4 | 0.12 | | | 1.0 | |
| NANO_5 | 1.2 | | 1.12 | infinity | |
| NANO_6 | 1.2 | | 20.0 | 0 | |
| NANO_7 | 100 | - | 20.0 | - | 0.015 – 0.25 |
| NANO_8 | 10 | - | 1.0 | - | 0.25 – 15.00 |

**Table 3.** Characteristics of the ICE70 aerosol particles. The unit of concentrations is cm$^{-3}$. Cases without "*" use aerosol size distributions (ASDs) from SPEC UAE project (Wehbe et al. 2021); cases marked with "*" use ASDs from Tessendort et al. (2021). The size ranges for M1, M2 and M3 are indicated in Figure 2c using different line patterns. The hygroscopicity parameters and size ranges are related to modes M1, M2 and M3, respectively.

| Name | Number concentration M1, M2 and M3 modes | Chemical composition of modes | Hygroscopicity ($\kappa$) | r range (µm) |
|---|---|---|---|---|
| ICE70_1 | 910.0 | KCl | 0.99 | 0.04 – 0.40 |
| | 0.07 | KCl – CaCl$_2$ mix | 0.74 | 0.40 – 0.75 |
| | 1.8 | CaCl$_2$ | 0.48 | 0.75 – 1.75 |
| ICE70_2 | 91.0 | KCl | 0.99 | 0.04 – 0.40 |
| | 0.007 | KCl – CaCl$_2$ mix | 0.74 | 0.40 – 0.75 |
| | 0.18 | CaCl$_2$ | 0.48 | 0.75 – 1.75 |
| ICE70_3* | 2010.0 | KCl | 0.99 | 0.15 – 0.40 |
| | 16.0 | KCl – CaCl$_2$ mix | 0.74 | 0.40 – 0.75 |
| | 6.0 | CaCl$_2$ | 0.48 | 0.75 – 5.40 |
| ICE70_4* | 201.0 | KCl | 0.99 | 0.15 – 0.40 |
| | 1.60 | KCl – CaCl$_2$ mix | 0.74 | 0.40 – 0.75 |
| | 0.60 | CaCl$_2$ | 0.48 | 0.75 – 5.40 |

**Table 4.** Characteristics of the NCM aerosol particles. The unit of concentrations is cm$^{-3}$.

| Name | Number concentration | Chemical composition | Hygroscopicity ($\kappa$) | r range (µm) |
|---|---|---|---|---|
| NCM_1 | 2100.0 | 85% KCl and 15 % NaCl | 1.00 | 0.05 – 0.15 |
| NCM_2 | 210.0 | 85% KCl and 15 % NaCl | 1.00 | 0.05 – 0.15 |