# Peer review of "Impact of hygroscopic seeding on the initiation of precipitation formation: results of a hybrid bin microphysics parcel model"

_Atmospheric Chemistry and Physics, 2021_

## Referee Comment (RC2)

**Review of ACP-2021-506**

In this manuscript, the authors used a cloud parcel model to investigate the characteristics of cloud droplet spectral evolution by condensation and collision and coalescence for various background CCN distributions and different updraft conditions. Then the impact of hygroscopic seeding material on cloud droplet spectral broadness was examined. The model they used seemed very appropriate for calculating cloud droplet growth processes in an adiabatic cloud parcel, limiting numerical diffusion by adopting moving bin boundaries for calculating condensation processes. The limitation was that this model did not take into account the entrainment and mixing processes, which certainly affect cloud droplet growth processes and droplet distributions in real clouds. However, for examining cloud droplet spectral broadening at earlier stages of cloud development, such limitation may be tolerable. The described impact of hygroscopic seeding material seems somewhat expected. Certainly seeding effect would be pronounced when seeding particles are big and such effect would be diminished when background CCN include many big particles. The scientific contribution of this manuscript mainly comes from the development of hybrid bin scheme that can be used for calculating condensational growth process without numerical diffusion. I think that this manuscript deserves publication in ACP after minor revision, addressing the comments I made below.

**Major comments:**

It is good to see that the Ostwald-ripening (OR) effect on droplet spectral broadening can also be significant under non-oscillating vertical velocity conditions. In a strict sense, however, what was presented in this manuscript was not exactly the same as the OR effect described in Yang et al. (2018), where the spectral broadening occurred since larger droplets grew but smaller droplets shrank. Such phenomenon can occur easily under oscillating vertical velocity condition: during updraft all droplets can grow but during downdraft larger droplets can still grow but smaller droplets may evaporate as they may become deactivated. In this manuscript, vertical velocity was always positive (updraft), although the value itself varied. So all activated droplets grew throughout the ascent regardless of their sizes but the important point was that the radius growth rate of larger droplets could be higher than that of smaller droplets near cloud base altitudes especially under low updraft conditions, resulting in broadening of

the cloud droplet distribution. Such spectral broadening can also be called the OR effect but the subtle difference from Yang et al. (2018) should be noted. In fact, the characteristics of spectral broadness of droplets that are grown by condensation under different CCN and updraft conditions were extensively examined by Yum and Hudson (Atmospheric Research, 2005), which clearly explained with cloud parcel model calculation and theoretical assessment that it was the differences between the ambient (cloud) supersaturation and the equilibrium supersaturations of different size droplets that determine spectral broadness of condensationally grown droplets: at lower ambient supersaturation, the differences between the ambient supersaturation and the equilibrium supersaturations of different size droplets are relatively larger than those at higher ambient supersaturation, and therefore broader spectra. Yum and Hudson (2005) should be cited when discussing the dependence of spectral broadening on supersaturation.

The description of Eq (1) is a little confusing. The indices i and k appear together for m and D. Does it mean that there exist multiple k values for each droplet size bin boundary, i? According to Table 1, a specific kappa value is associated with a specific mode of aerosol particles. So I guess that a specific k value is associated only with a certain range of i values. This should be clearly stated.

Line 173: Are the temperature and vertical velocity profiles different for different aerosol conditions or are they given as initial conditions? Temperature in the cloud parcel may become slightly different for different initial aerosol conditions since latent heat release can be slightly different. But the vertical velocity profile should have been prescribed. This sentence can misleadingly indicate that vertical velocity profile can be affected by the given initial aerosol distribution. This may be so but I doubt that the model took that into account.

Line 211: What the model calculates is the adiabatic LWC in the sense that the model does not allow heat exchange and mixing of the outside air. However, this adiabatic LWC can be different for different updraft conditions because different supersaturation (indicating the amount of excess vapor remaining without being condensed) can be generated for different updraft conditions, as demonstrated in this manuscript. What the authors indicate in this sentence is the maximum adiabatic LWC that can be obtained in the pseudo-adiabatic process which assumes that all excess water vapor is condensed and just saturation is maintained

during the ascent. Make it clear.

Line 261: It is stated that seeding has no significant effect on the growth rate of the drops formed on background aerosol particles. I would guess that adding seeding material would increase total droplet concentration and decrease the supersaturation, leading to broader spectra even only for the droplets formed on background aerosols. What were the change or difference of total droplet concentration and supersaturation caused by seeding?

Line 339: Background CCN concentration does not decrease. The number of activated cloud drops from background CCN may decrease. Rewrite the sentence.

In all size distribution plots, y-axis label is written as $N/dlog(r)$, not $dN/dlog(r)$. Are you sure? Then what does N mean here?

Figure 3: What do closed circles mean? No explanation is given in caption or in the text.

Figure 4: Integration of droplet size distribution would produce total droplet concentration. If I do that for the two droplet size distribution for two different updraft shown in Fig. 4a, I would find that the total droplet concentration is higher for the lower updraft. The y-axis is in log scale. So the actual difference of the concentrations might not be as dramatic as shown in the plot but it should still be true that the concentration is higher for the lower updraft. I do not understand this.

Figure 8: E is not clearly defined in the caption. Make it clear.

Table 4: What is NCM? No explanation in caption or in the text.

**Minor comments:**

L178: rewrite $w_1$.

L212: vapor flux → vapor surplus

L238: Move "(Wehbe et al., 2021)" to the end of the previous sentence.

L251: remove 'of' in front of *bg*.

---

## Author Comment (AC1)

Response to Reviewer 1.

*We greatly appreciate the careful review and constructive comments from the reviewer. We agree with the reviewer on most comments and tried very hard to address all the concerns in the responses and in the revised manuscript. A detailed description is added about the estimation of the errors caused by using first order Taylor series for the evaluation of the curvature and solution terms. We also show that the Hc parameter can be derived from first principles.*

*In this document, the original comments are in blue font and our responses are in italic type and black, furthermore the suggested modifications in the manuscript are in normal type and black.*

The authors develop a hybrid bin microphysical parcel model to study the impact of hygroscopic seeding on the precipitation formation. Moving bins and condensational growth of droplets with solute effect are used below and close to the cloud base. Fixed bins are applied at 100 m above the cloud base, where solute effect is ignored but collision coalescence is considered. Initial background aerosol conditions and updraft profiles for control cases are obtained from two weather modification field campaigns. For comparison, three types of seeding materials with different concentrations, sizes, and hygroscopicity are used to test how hygroscopic seeding might affect the precipitation formation. Although the hybrid parcel model is suitable and the observational data is valuable, results and conclusions from different model setups (e.g., aerosol size distribution, concentration, size, compostion of background aerosol and seeding materials, vertical velocity) are not clearly described. More efforts are needed to make the manuscript clear and convincing. I suggest major revision to improve the quality of the manuscript and my comments are listed below.

1. Figure 1. Is temperature also prescribed for the parcel model? If so, the parcel model used in this study is not adiabatic, which might explain why the ratios of liquid water content are large for some cases in Figure 8a (see my comment 13).

*The temperature profile is not prescribed, it is evaluated by calculating the adiabatic expansion of the parcel and the releasing latent heat of condensation. Because the calculation of the vapor uptake by the wet aerosol particles starts below the cloud base, the impact of the latent heat release of condensation is taken into consideration under the cloud base as well. If the ratio of the $Q_{seed}/Q_{ctrl}$ is larger than 1 the liquid water content (LWC) is super-adiabatic. In the control cases the LWC is close to the adiabatic value. Adiabatic LWC is defined as zero before the parcel reaches saturation, and the supersaturation is zero above the cloud base. At this level, the adiabatic LWC is very small around 0.1 g kg$^{-1}$. Therefore, even small changes in the sub-cloud aerosol conditions can result in large relative impact on the condensation. Superadiabatic LWC near the cloud base has been observed in some field projects (e.g. Blyth and Latham, 1985, Yum and Hudson, 2001).*

*To clarify the content of the Figure 1. we modify the figure caption:*

"Figure 1: The simulated temperature profiles and the prescribed updraft profiles in the numerical simulation for (a) the SPEC UAE cases and (b) the QCSRP cases. The background aerosol particles are BGUAE_1 and BGQNC_1, seeding materials are ICE70_2 and ICE70_4 in the figure (a) and (b), respectively. The plotted temperature profiles are calculated at the updraft profile of w$_3$ for each panel. The horizontal blue lines denote the altitude of the cloud base. The three horizontal black lines from bottom to top denote the level where the calculation

of the diffusional growth starts (RH = 70%), the levels 100 m and 1000 m above the cloud base."

*To clarify the content of the Fig. 8a, the following texts are inserted at line 335 in the original and at line 371 in the revised manuscript:*

"If the ratio of the $Q_{seed}/Q_{ctrl}$ is larger than 1, the LWC is superadiabatic, because in the control cases, the LWC is close to the adiabatic value at 100 m above the cloud base. The uptake of vapor by highly hygroscopic particles under the cloud base, especially in the case with a weak updraft, increases the LWC to above the adiabatic value. The amount of the surplus is small, so the effect is significant only when the adiabatic LWC is small (e.g., near the cloud base) and becomes negligible at a higher level above the cloud base. We hypothesize that the efficient vapor uptake by the coarse-mode hygroscopic particles can partly explain the observed superadiabatic LWC near the cloud base with weak updrafts in some field campaigns (Blyth and Latham, 1985, Yum and Hudson, 2001)."

*Further explanation for the content of Fig. 8a can be found in the original version of manuscript at lines 332- 335 and at lines 369- 371 in the revised manuscript:*

"The strong correlation reveals that the enhancement of the $H_c$ parameter due to the injection of highly soluble coarse particles (in this case, the number concentration of activated particles only slightly increased) can significantly increase the water vapor uptake under the cloud base (see the intense growth rate of the drops formed on nanoparticles in Figs. 6b and d)."

2. Eq. 1 approximates the Kappa-Kohler theory. The approximation might not hold for coarse-mode aerosols. I suggest using the complete Kappa-Kohler theory for the saturation ratio over an aqueous solution droplet (Eq. 6 in Petters and Kreidenweis 2007). If not, at least justification is needed.

*We agree that both the curvature and solution terms in Eq. 1.are simplified by using first order Taylor series to evaluate these terms. We justify that the approximation results in only small error. To prove this we calculate the following terms at different growth factors ($GF=D_d/D_{aer}$) and different aerosol sizes:*

$$T_1 = \frac{1}{1 + \kappa \dfrac{1}{GF^3 - 1}} \exp\left(\frac{A}{D_{aer} GF}\right)$$

$$T_2 = 1 + \frac{A}{D_{aer} GF} - \frac{\kappa}{GF^3 - 1}$$

$$REL\_ERR = \frac{T_1 - T_2}{T_1}$$

*The REL_ERR (relative errors) as a function of GF at discrete aerosol sizes are plotted in Fig. R1. The smallest value of the GF is 1.5. (At the beginning of the calculation of the condensational growth, the diameter of the haze particle is assumed 1.5 times of the dry aerosol particles.) The plots reveal that the relative errors are less than 20% even in the case*

*of the aerosol particles with diameter of 5 µm at the initial moment, and the error decreases steeply as the GF increases. When GF is greater than 2.0, the relative error is less than 2 % independent of the aerosol size. Using Eq. 6. in Petters and Kreidenweis 2007, we need to assume that the drop is at equilibrium with its environment, i.e., the diffusional growth rate of the drop is equal to zero. The disadvantage of using Eq. 6. in Petters and Kreidenweis 2007 is that it significantly overestimates the growth rate of the water drops formed on larger hygroscopic particles (e.g. coarse particles), because in the case of the larger hygroscopic particles it takes significantly longer time than the model time step to reach the equilibrium size even in the slightly undersaturated environment (see Fig. 7a in the manuscript). The reason of this is that neglecting the temperature difference between the drops and its environment results in large error for all sizes of drops and CCN if the condensation/evaporation rate is significant (Pruppacher and Klett, 1978).*

[Figure]

*Figure R1. The relative errors between the first order Taylor series and the complete form of the curvature and solution terms as a function of growth factor for aerosol diameters of 0.01 (black), 0.1 (red) and 5 microns (blue).*

*Sentences were added at the line 88 in the original and at lines 90 - 93 in the revised manuscript:*

"Both the curvature and solution terms (e.g. in Petters and Kreidenweis 2007) are approximated by the first order Taylor series in Eq. 1. The error caused by this approximation steeply decreases with the increase of the drop size. It is less than 20 % if the drop diameter is equal to the 1.5 times of the aerosol diameter, and less than 2 % if the ratio of the drop diameter and the aerosol diameter is greater than 2, even in the case of aerosol diameter of 5 µm."

3. Eq. 3 and Eq.6. The "new" parameters are confusing, e.g., it looks like that doubling the hygroscopicity parameter is equivalent to half coarse particles, which is not true. In my point

of view, "Hc" is not a general parameter that can be used for future applications, because it is not derived from the first principle. I suggest the authors add more justifications and discussions of why and the benefit to introduce this new parameter.

*We intend to introduce a dimensionless, bulk parameter that can be used to describe the impact of the coarse particles on the broadening of the drop spectrum and can be easily derived from both observational data and output of the numerical simulation. We agree that, in general, changes in $H_c$ by doubling the hygroscopicity of the coarse particles is not necessarily equivalent to the effect by reducing the concentration by half. However, we justify in the following that this parameter can be used to estimate the role of the coarse particles in the broadening of the DSD near the cloud base.*

*Let's start with the equation that describes the diffusional growth of water drops.*

$$\frac{dm_d}{dt} = 2\pi D_d \frac{\left[ S - 1 - \frac{4\sigma_{wa}}{R_v T \rho_w D_d} + \kappa \frac{m_a}{m_d - m_a} \frac{\rho_w}{\rho_a} \right] f_v}{\frac{L_v}{k_a^*}\left( \frac{L_v}{R_v T} - 1 \right) + \frac{R_v}{e_{sat,w} D_v^*}}$$

*For the diffusional growth of the drop formed on coarse particles, we assume that the solute term is dominant in the numerator of the above equation. Near the cloud base this is an acceptable approximation because of the large size of the coarse particles and the high concentration of the soluble components inside the droplets.*

*Neglecting the terms about the supersaturation and Kelvin effect, we get the following differential equation which is easy to solve analytically:*

$$\frac{dm_d}{dt} \approx C \cdot \kappa \cdot m_d^{-2/3}$$

*The solution of this equation is:*

$$m_d = \left( m_{d,0}^{5/2} + C \cdot \kappa \cdot t \right)^{2/5} \approx \left( C \cdot \kappa \cdot t \right)^{2/5}$$

*From this we can evaluate the reflectivity assuming monodisperse size distribution for coarse particles:*

$$Z \approx N m_d^2 \approx N \left( C \cdot \kappa \cdot t \right)^{4/5} \approx N \kappa \cdot t$$

*Because the lg(Z) is plotted in Fig. 3a, we can take the logarithmic of the above equation.*

$$\lg(Z) \approx \lg(N\kappa t)$$

*The impact of the coarse particles is normalized with the number concentration of the activated aerosol particles from the observations. The numerical simulation shows that the*

*efficiency of precipitation is negatively correlated to the number concentration of the activated aerosol particles. We can modify the equation above by taking into consideration of the number concentration of the activated aerosol particles ($n_{act}$)*

$$\lg(Z) \approx \lg\left(\frac{N\kappa t}{n_{act}}\right) + \lg(n_{act}) = \lg(H_c) + \lg(n_{act}) + \lg(t)$$

*The relation between the $H_c$ parameter and reflectivity can be presented more clearly if we change the x axis scale from linear to logarithmic in Fig. 3a (see Fig. R2). The plot can be interpreted as follows: If the value of $H_c$ is small, the role of the coarse particle comparing to that of all CCN particles is small in the formation of the larger drops. If the $H_c$ is above the threshold value of about $10^{-4}$ , the logarithm of reflectivity is approximately a linear function of the logarithm of the $H_c$ parameter. This relation reveals that the approximations we used are reasonable.*

[Figure]

*Figure R2. Same as Fig. 3 in the original manuscript but using the logarithmic scale in the horizontal axis for Hc.*

*To clarify this point, we modify the text at line 187 in the original and at line 204 in the revised manuscript:*

"If the value of $H_c$ is small, the impact of the coarse particles comparing to all of CCN particles is small in the precipitation formation. This can be interpreted as – is such case – coarse particles contribute insignificantly to the evolution of the DSD when compared to the fine particles. If $H_c$ is above of the threshold of about $10^{-4}$, the logarithm of the reflectivity is approximately a linear function of the $H_c$ parameter logarithm."

4. Line 164. "kappa  = 20 is the hygroscopicity of dry nanoparticles". Any reference to support such high value of hygroscopicity? I cannot find it in Tai et al., 2017.

*Based on the published data, we found that the vapor uptake of the NaCl-TiO₂ nanoparticles becomes efficient at a low relative humidity of 50 – 60 %, while the NaCl particles start to deliquesce above 70%.  The hygroscopicity of the nanoparticles is not constant, the observations show that it depends on the size of drops formed on them, and it is rather difficult to measure its value. The observations only reveal that about 5 to 10  times more mass of vapor is uptaken by the nano particles in the undersaturated environment if the diffusional growth of the drops occur in a sub-saturated environment within a reasonable*

*time (Bermeo et al, 2020). In the numerical studies, arbitrary values as high as 20 (e.g. Lompar et al, 2018) were used for the hygroscopicity. Contrary to the above cited research, we considered the dependence of the hygroscopicity on the GF (see Eq. 5 in the manuscript). Using this equation, the hygroscopicity is less sensitive to the initial value, and it is more sensitive to X. If X = 1 and GF = 5, the hygroscopicity of the nano particles is less than twice of the hygorscopicity of the salt if $\kappa_0$ is equal to or less than 20.*

*We clarify this point in the manuscript. Explanation is added at line 166 in the original and at line 173 in the revised manuscript:*

"There are no available observation data to calibrate the value of $\kappa_0$. In the numerical studies, arbitrary values as high as 20 (e.g. Lompar et al, 2018) were used for the hygroscopicity of the nanoparticles. Contrary to the above cited research, the dependence of hygroscopicity on the GF is considered in this study. If $X$ is set to 0.1 (NANO_2) and 1.0 (NANO_1), the nanoparticles uptake about 5 and 2 times, respectively, more vapor than the salt particles (NANO_5) during the air parcel ascends from the levels of the RH = 70% to the cloud base. This enhancement of the vapor uptake agrees well with the laboratory observation data which shows that about 5 to 10 times more mass of vapor is taken up by the nanoparticles in the sub-saturated environment within a reasonable time (Bermeo et al, 2020)."

ref:

Bermeo, M., Hadri, N. E., Ravaux, F., Zaki, A., Zou, L., Jouiad M., 2020: Adsorption Capacities of Hygroscopic Materials Based on NaCl-TiO 2 and NaCl-SiO 2 Core/Shell Particles. Journal of Nanotechnology **2020,** *2020* , 1-16.
https://doi.org/10.1155/2020/3683629

Lompar, M., Ćurić, M., Romanic, D., Zou, L., Liang, H., 2018: Precipitation enhancement by cloud seeding using the shell structured TiO2/NaCl aerosol as revealed by new model for cloud seeding experiments. Atmos. Res. 212, 202-212

5. Lines 187-189: "The monotonic increase of the reflectivity with increasing Hc…" This statement is not convincing. To support this argument, the authors need to do sensitivity studies by just changing the hygroscopicity of the coarse particles but keep other variables (e.g., droplet size distribution, aerosol number concentration, and vertical velocity) the same.

*We did not intend to emphasize the separate impact of the hygroscopicity and number concentration, we intended to consider the impact of the coarse particles relative to all CCN.*

*We modified the text at line 204 in the revised manuscript:*

"If the value of $H_c$ is small, the impact of the coarse particles comparing to all CCN particles is small in the precipitation formation. This can be interpreted as – in such case –coarse particles contribute insignificantly to the evolution of the DSD when compared to the fine particles. If $H_c$ is above a threshold of about $10^{-4}$, the logarithm of the reflectivity is approximately a linear function of the $H_c$ parameter logarithm."

6. Figure 4a. Sub-micrometer droplets (r<1 um) exist at 100 m above the cloud base, but they do not exist at 1000 m above the cloud base. Please explain why? Are they all activated as cloud droplets?

*The smallest drop size plotted in the figure is larger than critical drop radii for the drops formed on the fine aerosol particles (0.13 μm and 0.06 μm for the updrafts of 1 ms$^{-1}$ and 5 ms$^{-1}$, respectively). So all drops in the plotted size range grow through the vapor condensation above this level.*

*Sentences to clarify this point are inserted to the text at line 204 in the original and at line 224 in the revised manuscript:*

"Figure 4a shows the DSDs at 100 m and 1 km above the cloud base. Note that the critical droplet radii for both updrafts (0.13 μm and 0.06 μm for the updrafts of 1 ms$^{-1}$ and 5 ms$^{-1}$, respectively) are smaller than the minimum droplet size plotted in the figure, the concentration of the drops larger than the critical size actually remain constant during the ascent of the parcel. "

7. Figure 4c. Droplet radius for r0=5 um is already over 10 um at the beginning of the simulation. How do you calculate the initial droplet radius at the beginning of the simulation?

*The plots in Fig. 4c start at the altitude of 3800 m, and the calculation of the diffusional growth of water drops starts at 3300 m (see Fig. 1c). The reason that we plot only a subrange of the whole simulated range, is that we intend to take a close look near the cloud base.*

*A sentence is added to the figure caption to clarify this:*

"Note, the plots in Fig. 4c start at the altitude of 3800 m, and the calculation of the diffusional growth of the water drops starts at 3300 m (see Fig. 1c)."

*To avoid the division by zero in the solution term in Eq. 1, the integration of the equation starts with the haze particle having 1.5 times diameter of the dry aerosol particle.  A sentence is added to clarify this at line 88 in the original and at line 93 in the revised manuscript:*

"To avoid the division by zero at the evaluation of the solution term in eq 1. at the beginning of the calculation of the condensational growth, the diameter of the haze particle is assumed 1.5 times of the dry aerosol particle."

8. Line 206. "Figure 4c shows the evolution of drop sizes with different initial dry radii." I'm confused. Are monodisperse aerosols used for Figure 4c and 4d?

*Sorry for the confusion. No, we did not use monodisperse aerosols for these plots. We intend to plot the evolution of the water drops at discrete sizes which represent the spectrum of the aerosol particles.*

*A sentence is added to clarify this at line 207 in the original and at line 230 in the revised manuscript.*

"Figure 4c shows the evolution of drop sizes with different initial dry radii. These discrete initial radii are chosen to represent the diffusional growth of drops formed on aerosol particles with different sizes and hygroscopicities."

9. Lines 258-261. "the broadening effect is found to be negatively associated with the concentration of the background coarse particles…" Text here might mislead the reader that seeding is more efficient when the concentration of the background coarse particles is less. It is unfair to make comparisons between BGQNC_1 and BGQNM_1 because they have different aerosol size distributions, number concentrations, and compositions.

*That was what we intended to conclude based on the aerosol backgrounds assessed. The main purpose of this research is to study the efficiency of the seeding in different environmental conditions, which means different background aerosol size distributions and chemical compositions as well. See the purpose of the research at line 63.*

*Numbers of papers have been published on how natural precipitation formation depends on the number concentration and/or the size distribution of the background aerosol particles (typical examples are the papers comparing the precipitation formation in polluted and clean air mass).*

*The importance of this comparison study is also supported by the conclusion of the field experiment (Wehbe et al, 2021):*

*„Similarly, the modelling work of Segal et al. (2004) indicates a decrease in seeding effects in the presence of large background CCN due to their efficient collision. Hence, given the comparable sizes between the existing GCCN over the UAE and typical seeding particles, it is unclear if hygroscopic seeding can be effective in these clouds. Modelling studies are needed to investigate whether the concentration and hygroscopicity of the background GCCN are high enough to cause a natural competition effect. Furthermore, modelling studies can help assess the effectiveness of perhaps larger seeding particle sizes (10–15 μm) in augmenting this potentially active natural competition effect and/or in initiating C-C."*

*ref:*

*Segal, Y., Khain, A., Pinsky, M., and Rosenfeld, D.: Effects of hygroscopic seeding on raindrop formation as seen from simulations using a 2000-bin spectral cloud parcel model, Atmospheric Research, 71, 3-34, 2004.*

*Wehbe, Y., Tessendorf, S. A., Weeks, C., Bruintjes, R., Xue, L., Rasmussen R. M., Lawson, P., Woods, S., and Temimi, M. 2021: Analysis of aerosol-cloud interactions and their implications for precipitation formation using aircraft observations over the United Arab Emirates, Atmospheric Chemistry and Physics, https://doi.org/10.5194/acp-21-12543-2021*

*To emphasize our aim in this study more clearly, we modify the text in the paragraph about the purpose of this research at line 65:*

(i) "Understanding the mechanisms of the spectral broadening induced by characteristics of both background aerosols and hygroscopic seeding materials. A large number of numerical experiments are performed to examine the dependence of seeding efficiency on the type of seeding materials, on the cloud dynamics, and on the characteristics of background aerosols. ==The assessment of the broadening of drop spectra at different CCN concentrations and different characteristics of the coarse particles may advance the designation of the optimal environmental conditions for the hygroscopic seeding.== We hypothesize that the competition for the available vapor near the cloud base between the drops formed on fine-mode ($r < 0.1\ \mu m$) and coarse-mode particles ($r > 1\ \mu m$) impacts the precipitation formation. "

10. Figure 8. What are the reasons for the cases with high ratio of liquid water contents (up to 1.6)? I would expect the liquid water content at 100 m above cloud base is similar (close to adiabatic value) for different cases, as shown in Figure 4b.

*If the ratio of the $Q_{seed}/Q_{ctrl}$ is larger than 1 the liquid water content (LWC) is super-adiabatic. In the control cases the LWC is close to the adiabatic value. (At this level, the adiabatic LWC is very small around 0.1 g kg$^{-1}$.) Therefore even small changes in the sub-cloud aerosol conditions can result in large relative impact on the condensation. Super-adiabatic LWC near the cloud base has been observed in some field projects (e.g. Blyth and Latham, 1985, Yum and Hudson, 2001).*

*To clarify the content of Fig 8a, the texts are inserted at line 335 in the original and at line 371 in the revised manuscript:*

"If the ratio of the $Q_{seed}/Q_{ctrl}$ is larger than 1 the LWC is superadiabatic, because in the control cases, the LWC is close to the adiabatic value at 100 m above the cloud base. The uptake of vapor by highly hygroscopic particles under the cloud base, especially in the case with a weak updraft, increases the LWC to above the adiabatic value. The amount of the surplus is small, so the effect is significant only when the adiabatic LWC is small (e.g., near the cloud base) and becomes negligible at a higher level above the cloud base. We hypothesize that the efficient vapor uptake by the coarse-mode hygroscopic particles can partly explain the observed superadiabatic LWC near the cloud base with weak updrafts in some field campaigns (Blyth and Latham, 1985, Yum and Hudson, 2001)."

*Further explanation for the content of Fig. 8a can be found in the original version of manuscript at lines 332- 335 and at lines 369- 371 in the revised manuscript:*

"The strong correlation reveals that the enhancement of the $H_c$ parameter due to the injection of highly soluble coarse particles (in this case, the number concentration of activated particles only slightly increased) can significantly increase the water vapor uptake under the cloud base (see the intense growth rate of the drops formed on nanoparticles in Figs. 6b and d)."

Minor comments:

1. Line 105: Add more descriptions of bin method, e.g., gridded uniformly in radius, or mass.

 *A sentence is added and modified at line 104 in the original version of manuscript and at line 111 in the revised manuscript about the moving bin method:*

"The initial size distributions of the aerosol particles in different categories are divided into 70 mass bins ranging from radius of 0.016 μm to 46.6 μm with bin mass increment factor of the square root of 2."

*The sentence at line 105 in the original version of manuscript and at line 112 in the revised manuscript is modified as follows:*

In the case of the fixed bin scheme, 48 mass doubling bins are defined over the radius range from 0.1 μm to 5.0 mm.

2. Line 155: Please add more descriptions of the seeding materials (nanoparticles, ICE70, and NCM).

*We add more information about the nano particles at line 173 in the revised manuscript:*

*See our responses to point 4.*

*Sentence at line 167 in the original version of manuscript and at line 181 in the revised manuscript is inserted for the ICE70 particles:*

"This type of burning flares produce both fine- and coarse-mode hygroscopic particles, but the concentration of the fine particles is about two-order of magnitude higher than that of the coarse particles (Bruintjes et al., 2012)."

*To our knowledge there is no publication about the NCM seeding materials. Sentence at line 171-172 in the original version of manuscript and at lines 187-189 in the revised manuscript is modified to add more information about the NCM particles:*

"The NCM seeding material (Table 4 and Fig. 2c) is released from hygroscopic flares and was recently developed by the National Center of Meteorology in UAE. Comparing to the ICE70 particles, the size distributions of the NCM particles are narrower, it consists of only fine particles. "

*ref:*

*Bruintjes, R.T., Salazar, V., Semeniuk, T. A., Buseck, P., Breed, D.W., and Gunkelman, J., 2012: Evaluation of Hygroscopic Cloud Seeding Flares. Journal of Weather Modification. Vol. 44 No. 1*

3. Figure 1. Solid circles are cases for BGUAE_1? It is not clear in the Figure and captions.

*We think you probably mean Figure 3.*

*The solid circles represent all of the six other background cases (BGUEA_1, BGUAE_3, BGUAE_4, BGUAE_5, BGUAE_6 and BGQNC_2 with three corresponding updraft profiles). Because we changed the x axis to the logarithm scale, the dots belong to the case BGUAE_4 cannot be plotted ($H_c = 0$ in this case). A sentence is added to the figure caption and one sentence is inserted in the text to clarify this.*

*The figure caption is modified:*
"Figure 3: Reflectivity calculated from the modelled DSD (a) 100 m and (b) 1000 m above the cloud base for non-seeded control cases. Symbols with different colours represent different updraft velocities at the cloud base as indicated by the key. Three different background aerosol size distributions are presented by solid triangles, squares, and open circles, respectively (see the legend in panel a.). The solid circles represent all of the six other background conditions (BGUEA_1, BGUAE_3, BGUAE_5, BGUAE_6 and BGQNC_2 with three corresponding updraft profiles except for BGUAE_4 case where $H_c$ is zero)."

*The following sentence is inserted at the line 193 in the original version of manuscript and at line 210 in the revised manuscript:*
"(all the other background cases are not distinguished from each other by using different symbols, and they are denoted by closed circle)"

4. Line 184. "27 control cases are simulated. Figure 3a…" Not all 27 control cases are shown in Figure 3a. Add more discussions of which cases are plotted in Fig. 3.

*All cases are plotted, but some points are overlapped.  If you zoom in, you can count almost all 27 cases, but one. Explanation is added in the figure caption (see our response above). Because we changed the scale to logarithmic in Figs. 3a and b, the BGUAE_4 case cannot be plotted because the $H_c$ is zero in this case.*

5. Figure 4a and others (e.g., Figure 5). The y label should be "dN/dlog(r)". "d" is missing.

*Thanks, the labels are modified.*

The following references are added to the manuscript:

Blyth, A.M., Latham, J., 1985. An airborne study of vertical structure and microphysical variability within a small cumulus. Q. J. R. Meteorol. Soc. 111, 773–792.

Yum, S.S., Hudson, J.G., 2001. Cloud microphysical relationships in warm clouds. Atmos. Res. 57, 81– 104.

Yum, S.S., Hudson, J.G., 2005: Adiabatic predictions and observations of cloud droplet spectral broadness. Atmos. Res. 73, 203– 223.

Bermeo, M., Hadri, N. E., Ravaux, F., Zaki,  A., Zou, L., Jouiad M., 2020: Adsorption Capacities of Hygroscopic Materials Based on NaCl-TiO 2 and NaCl-SiO 2 Core/Shell Particles. Journal of Nanotechnology **2020,** *2020* , 1-16. https://doi.org/10.1155/2020/3683629

Lompar, M., Ćurić, M., Romanic, D., Zou, L., Liang, H., 2018: Precipitation enhancement by cloud seeding using the shell structured TiO2/NaCl aerosol as revealed by new model for cloud seeding experiments. Atmos. Res. 212, 202-212

---

## Author Comment (AC2)

Response to Reviewer 2.

*We greatly appreciate the careful review and constructive comments from the reviewer. We agree with the reviewer on most comments and tried very hard to address all the concerns in the responses and in the revised manuscript. More detailed descriptions were added about the equations used to calculate the diffusional growth. We clarified the description about the impact of seeding on the evolution of the water drops formed on background aerosol particles. The meaning of Ostwald ripening effect used in this study was also explained.*

*In this document, the original comments are in blue font and our responses are in italic type and black, furthermore the suggested modifications in the manuscript are in normal type and black.*

In this manuscript, the authors used a cloud parcel model to investigate the characteristics of cloud droplet spectral evolution by condensation and collision and coalescence for various background CCN distributions and different updraft conditions. Then the impact of hygroscopic seeding material on cloud droplet spectral broadness was examined. The model they used seemed very appropriate for calculating cloud droplet growth processes in an adiabatic cloud parcel, limiting numerical diffusion by adopting moving bin boundaries for calculating condensation processes. The limitation was that this model did not take into account the entrainment and mixing processes, which certainly affect cloud droplet growth processes and droplet distributions in real clouds. However, for examining cloud droplet spectral broadening at earlier stages of cloud development, such limitation may be tolerable. The described impact of hygroscopic seeding material seems somewhat expected. Certainly seeding effect would be pronounced when seeding particles are big and such effect would be diminished when background CCN include many big particles. The scientific contribution of this manuscript mainly comes from the development of hybrid bin scheme that can be used for calculating condensational growth process without numerical diffusion. I think that this manuscript deserves publication in ACP after minor revision, addressing the comments I made below.

Major comments:

It is good to see that the Ostwald-ripening (OR) effect on droplet spectral broadening can also be significant under non-oscillating vertical velocity conditions. In a strict sense, however, what was presented in this manuscript was not exactly the same as the OR effect described in Yang et al. (2018), where the spectral broadening occurred since larger droplets grew but smaller droplets shrank. Such phenomenon can occur easily under oscillating vertical velocity condition: during updraft all droplets can grow but during downdraft larger droplets can still grow but smaller droplets may evaporate as they may become deactivated. In this manuscript, vertical velocity was always positive (updraft), although the value itself varied. So all activated droplets grew throughout the ascent regardless of their sizes but the important point was that the radius growth rate of larger droplets could be higher than that of smaller droplets near cloud base altitudes especially under low updraft conditions, resulting in broadening of the cloud droplet distribution. Such spectral broadening can also be called the OR effect but the subtle difference from Yang et al. (2018) should be noted.

*A sentence is added to explain the difference between our interpretation of the OR and that of Yang et al. (2018) at line 198 in the original version of manuscript and at line 216 in the revised manuscript:*

"In this study, the Ostwald-ripening effect is subtly different from Yang et al. (2018). In Yang et al. (2018), the spectral broadening is a result of the shrinking of the smaller drops and growth of larger drops in an oscillating vertical velocity condition. In our study, the vertical velocity is always positive and all droplets larger than the critical size can growth throughout the ascent, but the larger drops grow faster than the smaller drops, resulting in broadening of the size distribution."

In fact, the characteristics of spectral broadness of droplets that are grown by condensation under different CCN and updraft conditions were extensively examined by Yum and Hudson (Atmospheric Research, 2005), which clearly explained with cloud parcel model calculation and theoretical assessment that it was the differences between the ambient (cloud) supersaturation and the equilibrium supersaturations of different size droplets that determine spectral broadness of condensationally grown droplets: at lower ambient supersaturation, the differences between the ambient supersaturation and the equilibrium supersaturations of different size droplets are relatively larger than those at higher ambient supersaturation, and therefore broader spectra. Yum and Hudson (2005) should be cited when discussing the dependence of spectral broadening on supersaturation.

*Thank you for drawing our attention to this paper. It is fully relevant.*

*We cite the paper in section 4.1 (line 218) and another paper published by these authors is also cited in section 5 (line 335).*

*We insert the following sentences at line 218 in the original version of manuscript and at line 246 in the revised manuscript:*

"Similar conclusions were reported by Yum and Hudson (2005). They numerically simulated the diffusional growth of water drops formed on aerosol particles with homogenous chemical composition (salt), and they also found that the curvature and solution terms could significantly impact the broadening of the DSD near the cloud base if the updraft velocity was small. "

*and the following text is inserted at line 335 in the original version of manuscript and at line 368 in the revised manuscript:*

"If the ratio of the $Q_{seed}/Q_{ctrl}$ is larger than 1 the LWC is superadiabatic, because in the control cases, the LWC is close to the adiabatic value at 100 m above the cloud base. The uptake of vapor by highly hygroscopic particles under the cloud base, especially in the case with a weak updraft, increases the LWC to above the adiabatic value. The amount of the surplus is small, so the effect is significant only when the adiabatic LWC is small (e.g., near the cloud base) and becomes negligible at a higher level above the cloud base. We hypothesize that the efficient vapor uptake by the coarse-mode hygroscopic particles can partly explain the observed superadiabatic LWC near the cloud base with weak updrafts in some field campaigns (Blyth and Latham, 1985, Yum and Hudson, 2001)."

The description of Eq (1) is a little confusing. The indices i and k appear together for m and D. Does it mean that there exist multiple k values for each droplet size bin boundary, i? According to Table 1, a specific kappa value is associated with a specific mode of aerosol particles. So I guess that a specific k value is associated only with a certain range of i values.

This should be clearly stated.

*The mass variables for the aerosol particles are given by one dimensional array, that means we use the same bin intervals for the different types of the aerosol particles. The size variables for the mass of drops and the diameter of water drops are described by two-dimensional arrays in this model. The first dimension is the size, and second dimension means the different (distinct) types of the aerosol particles the drops formed upon. So, we can simulate the diffusional growth of water drops separately even if the aerosol particles inside the drop have the same size but different hygroscopicities. In this study, the hygroscopicity is given in non-overlapping, separate size categories of the aerosol particles. This is the reason that we deleted the k index associated with aerosol mass in Eq. 1. However, even in this case, we need two-dimensional variables for the drops. Although at the start of the simulation there is no size overlapping among the aerosol particles at different hygroscopicities, size overlapping can occur at a later time as the drops formed on aerosol particles with smaller size but larger hygroscopicity can grow faster than the drops formed on aerosol particles with larger size and smaller hygroscopicity (e.g. in BGQNC_1 case).*

*Texts at line 84 is modified in the original version of the manuscript and at same line in the revised manuscript:*

"where $m_{d,i,k}$ and $D_{d,i,k}$ are two-dimensional variables of the mass and the diameter of the water drop, respectively. The index $i$ represents the bin boundary for the mass or size of the particles. The index $k$ represents the type of the aerosol particles on which drops formed. The size distributions of the different types of aerosol particles are given with the same mass bin intervals, the $m_{ap,i}$ is the mass of the aerosol particles at the $i^{th}$ bin boundary."

*The sentence at the line 95 in the original version of manuscript and at line 102 in the revised manuscript is modified:*

"At 100 m above the cloud base, the size of drops is large enough to neglect both the curvature and the solution terms, so the aerosol mass inside of the drops is not tracked, and Eq. 1 can be simplified to Eq. 2:"

Line 173: Are the temperature and vertical velocity profiles different for different aerosol conditions or are they given as initial conditions? Temperature in the cloud parcel may become slightly different for different initial aerosol conditions since latent heat release can be slightly different. But the vertical velocity profile should have been prescribed. This sentence can misleadingly indicate that vertical velocity profile can be affected by the given initial aerosol distribution. This may be so but I doubt that the model took that into account.

*We agree that we need to clarify the description of Fig. 1. The velocity profiles are prescribed, and they are not impacted by the latent heat release of condensation. However, the temperature profiles are calculated. We mentioned the types of the seeding materials in the text and the types of the background aerosol particles in the figure caption relevant for Figs. 1a and 1b. To better clarify this point, we add more information on the initial aerosol conditions in the caption of Fig. 1.*

*We modify the figure caption:*

"Figure 1: The simulated temperature profiles and the prescribed updraft profiles in the numerical simulation for (a) the SPEC UAE cases and (b) the QCSRP cases. The background

aerosol particles are BGUAE_1 and BGQNC_1, seeding materials are ICE70_2 and ICE70_4 in the figure (a) and (b), respectively. The plotted temperature profiles are calculated at the updraft profile of $w_3$ for each panel. The horizontal blue lines denote the altitude of the cloud base. The three horizontal black lines from bottom to top denote the level where the calculation of the diffusional growth starts (RH = 70%), the levels 100 m and 1000 m above the cloud base."

Line 211: What the model calculates is the adiabatic LWC in the sense that the model does not allow heat exchange and mixing of the outside air. However, this adiabatic LWC can be different for different updraft conditions because different supersaturation (indicating the amount of excess vapor remaining without being condensed) can be generated for different updraft conditions, as demonstrated in this manuscript. What the authors indicate in this sentence is the maximum adiabatic LWC that can be obtained in the pseudo-adiabatic process which assumes that all excess water vapor is condensed and just saturation is maintained during the ascent. Make it clear.

*We agree that we did not compose the sentence correctly. The liquid water content cannot be exactly equal to the adiabatic LWC, because the supersaturation is never equal to zero in these simulated cases even in the case of weak updraft. We modified the text to clarify this at line 210 in the original version of manuscript and at line 235 in the revised manuscript:*

"In the case of weak updrafts, the LWC approaches the adiabatic LWC immediately above the cloud base. (The LWC profile simulated with the weakest vertical velocity is closest to the adiabatic LWC profile. This difference is not discernible in Fig. 4b.) In the strong updraft (updraft profile of $w_1$) the vapor surplus due to the updraft exceeds the vapor depletion due to the condensation. Therefore, LWC approaches the adiabatic value only at a higher elevation, at about 100 m above the cloud base. Adiabatic LWC is defined as equal to zero before the parcel reaches saturation, and the supersaturation is zero above the cloud base."

Line 261: It is stated that seeding has no significant effect on the growth rate of the drops formed on background aerosol particles. I would guess that adding seeding material would increase total droplet concentration and decrease the supersaturation, leading to broader spectra even only for the droplets formed on background aerosols. What were the change or difference of total droplet concentration and supersaturation caused by seeding?

*The total amount of the condensed water is hardly changed by seeding, comparing to the control cases. Seeding particles grow by depleting water vapor, and at the same time, suppressing the growth of the background small droplets (Ostwald-ripening effect). When nanoparticles are used as seeding material, the maximum supersaturation was not impacted by seeding. When the flare particles are used for seeding (e.g., ICE70), the supersaturation is reduced. This reduction is consistent with the significant increase of the CCN due to seeding. However, even in this case, the total amount of condensed water vapor was the same in the control and seeded cases. Seeding results in a significant decrease of the water vapor uptake by the background aerosol particles (see line 308 in the manuscript). Because the average size of the flare particles is nearly the same as that of the background fine particles, they do not have the advantage of competing for more water vapor, so the size distribution remains narrow. To demonstrate this effect, the maximum supersaturation and the mass of the condensed water vapor in the control and seeded cases are added for each plotted case in Fig. 6. The impact of the seeding on the concentration of the activated background aerosol*

*particles in the case of the nanoparticles is not clearly described, so one sentence clarifying this is added to the text. Although the impact of the flare particles on the activation of background aerosols is mentioned in the original version of the manuscript (see lines 379 – 383 in the current version of the manuscript), sentences are inserted to make this effect clearer.*

*The following sentences are inserted in the manuscript at line 263 in the original version of manuscript and at line 293 in the revised manuscript:*

"Seeding by nanoparticles, due to their small concentration, hardly impacts the supersaturation above the cloud base and the amount of water vapor uptake by the background aerosol particles (see $s_{max,ctrl}$ and $s_{max,seed}$ as well as $Q_{bg,ctrl}$ and $Q_{bg,seed}$ in Fig. 6). In line with the small impact of the seeding on the supersaturation and the LWC the results of the numerical simulation show that the number concentration of the drops formed on background CCN does not change due to the seeding."

*and at line 308 in the original version of manuscript and at line 343 in the revised manuscript:*

"The doubling of the aerosol concentration due to seeding reduces the maximum supersaturation by 20% and halves the amount of the condensed water on the background aerosol particles (see $s_{max,ctrl}$ and $s_{max,seed}$ as well as $Q_{bg,ctrl}$ and $Q_{bg,seed}$ in Fig. 6e). The flare particles suppress the growth of the drops formed on the background aerosols, they mainly suppress the growth of drops formed on the fine-mode background aerosol particles and only slightly suppress the growth of the drops formed on the coarse-mode aerosols. However, their growth rate does not exceed significantly the growth rate of the background aerosol particles having the same size."

*Caption for Fig. 6 is also modified:*

"Figure 6: The vertical profiles of the growth rate of drops formed on background aerosol particles with different sizes and hygroscopicities (first column) and that formed on different types of seeding materials with different sizes and hygroscopicities (second column). In the left panels, the solid lines denote control cases, and the dashed lines denote seeded cases. Blue and black vertical lines denote the location of the cloud base and that of the maximum supersaturation. The $s_{max,ctrl}$ and $s_{max,seed}$ mean the calculated maximum supersaturation belong to the unseeded and seeded cases, respectively. The $Q_{bg,ctrl}$ and $Q_{bg,seed}$ mean the amount of the condensed water on the background aerosol particles in the control and seeded cases at 100 m above the cloud base, respectively. "

Line 339: Background CCN concentration does not decrease. The number of activated cloud drops from background CCN may decrease. Rewrite the sentence.
In all size distribution plots, y-axis label is written as N/dlog(r), not dN/dlog(r). Are you sure? Then what does N mean here?

*We assumed that the potential cloud condensation nuclei are called condensation nuclei (CN), and the activated condensation nuclei (subset of CN) are called CCN. However, we realize that this distinction is not generally applied, and we modify the sentence in line 339 in the original version of manuscript and at line 383 in the revised manuscript:*

"A notable decrease in the number concentration of activated drops from background CCN was only found in the case of a weak updraft (w = 1.0 m s$^{-1}$)."

*Sorry that the label text is wrong both in Figs. 4 and 5 (and Figs. 2 and 7 too). The correct label is dN/dlog(r). We modify the labels in these figures.*

Figure 3: What do closed circles mean? No explanation is given in caption or in the text.

*The solid circles represent all of the six other background cases (BGUEA_1, BGUAE_3, BGUAE_4, BGUAE_5, BGUAE_6 and BGQNC_2 with three, corresponding updraft profiles). Because we changed to the logarithm scale in the case of 'x' axis the dots belong to the case BGUAE_4 cannot be plotted ($H_c$ = 0 in this case). A sentence is added to the figure caption and one sentence is inserted in the text to clarify this.*

*The figure caption is modified:*
"Figure 3: Reflectivity calculated from the modelled DSD (a) 100 m and (b) 1000 m above the cloud base for non-seeded control cases. Symbols with different colours represent different updraft velocities at the cloud base as indicated by the key. Three different background aerosol size distributions are presented by solid triangles, squares, and open circles, respectively (see the legend in panel a.). The solid circles represent all of the six other background conditions (BGUEA_1, BGUAE_3, BGUAE_5, BGUAE_6 and BGQNC_2 with three corresponding updraft profiles, except BGUAE_4 case where $H_c$ is zero)."

*The following sentence is inserted at the line 193 in the original version of manuscript and at line 210 in the revised manuscript:*
"(all the other background cases are not distinguished from each other by using different symbols, and they are denoted by closed circle)"

Figure 4: Integration of droplet size distribution would produce total droplet concentration. If I do that for the two droplet size distribution for two different updraft shown in Fig. 4a, I would find that the total droplet concentration is higher for the lower updraft. The y-axis is in log scale. So the actual difference of the concentrations might not be as dramatic as shown in the plot but it should still be true that the concentration is higher for the lower updraft. I do not understand this.

*Unfortunately, the resolution of the plots is not good enough to distinguish the two curves at the peak.*

*This table shows the values of the functions belong to the two updraft profiles near the peak of the curves.*

| radius (μm) | dN/dlog(r) (m$^{-3}$) at $w_3$ | dN/dlog(r) (m$^{-3}$) at $w_1$ |
|---|---|---|
| 2.48 | $8.32 \times 10^8$ | $1.45 \times 10^8$ |
| 3.13 | $7.41 \times 10^9$ | $9.73 \times 10^9$ |
| 3.94 | $8.78 \times 10^8$ | $2.40 \times 10^8$ |

| | | |
|---|---|---|
| *4.96* | *5.29×10^7* | *2.37×10^7* |

*The data in the table shows that dN/dlog(r) at w₁ (strong updraft) is larger than that at w₃ only at the radius of 3.13 μm. This positive deviation at the peak is compensated by the negative deviation in other radii. We tried to zoom in Fig. 4a by starting the vertical scale at $10^4$. It helps a little bit.*

Figure 8: E is not clearly defined in the caption. Make it clear.

*We modified the figure caption:*

"Figure 8: (a) The impact of the change of the $H_c$ parameter on the vapor uptake. The vertical coordinate represents the ratio of liquid water contents belong to the seeded and associated control cases. (b) This panel shows how the broader size distribution generated by the Ostwald-ripening effect close to the cloud base ($E_{cb+100}$) can result in broadening of the size distribution comparing to the control case 1000 m above the cloud base ($E_{cb+1000}$). $E_{cb+100}$ and $E_{cb+1000}$ mean the calculated change of reflectivity due to seeding (Eq. 4) at 100 and 1000 m above the cloud base, respectively."

Table 4: What is NCM? No explanation in caption or in the text.

*To our knowledge there is no publication about the NCM seeding materials, sentence at lines 171-172 in the original version of manuscript and at lines 187 – 189 in the revised manuscript is modified to add more information about the NCM particles:*

"The NCM seeding material (Table 4 and Fig. 2c) is released from hygroscopic flare recently developed by the National Center of Meteorology at UAE. Comparing to the ICE70 particles, the size distributions of the NCM particles are narrower, and it consists of only fine particles."

**Minor comments:**
L178: rewrite w1.
*The text is modified.*

L212: vapor flux → vapor surplus
*The text is modified:*

L238: Move "(Wehbe et al., 2021)" to the end of the previous sentence.
*The text is modified.*

L251: remove 'of' in front of *bg*.
*The text is modified.*

The following references are added to the manuscript:

Blyth, A.M., Latham, J., 1985. An airborne study of vertical structure and microphysical variability within a small cumulus. Q. J. R. Meteorol. Soc. 111, 773–792.

Yum, S.S., Hudson, J.G., 2001. Cloud microphysical relationships in warm clouds. Atmos. Res. 57, 81– 104.

Yum, S.S., Hudson, J.G., 2005: Adiabatic predictions and observations of cloud droplet spectral broadness. Atmos. Res. 73, 203– 223.

Bermeo, M., Hadri, N. E., Ravaux, F., Zaki, A., Zou, L., Jouiad M., 2020: Adsorption Capacities of Hygroscopic Materials Based on NaCl-TiO 2 and NaCl-SiO 2 Core/Shell Particles. Journal of Nanotechnology **2020,** *2020* , 1-16.
https://doi.org/10.1155/2020/3683629

Lompar, M., Ćurić, M., Romanic, D., Zou, L., Liang, H., 2018: Precipitation enhancement by cloud seeding using the shell structured TiO2/NaCl aerosol as revealed by new model for cloud seeding experiments. Atmos. Res. 212, 202-212